# Retrieval of an Ice Water Path over the Ocean from ISMAR and MARSS millimeter/submillimeter brightness temperatures

Manfred Brath[1], Stuart Fox[2], Patrick Eriksson[3], R. Chawn Harlow[2], Martin Burgdorf[1], and Stefan A. Buehler[1]

[1]Meteorologisches Institut, Fachbereich Geowissenschaften, Centrum für Erdsystem- und Nachhaltigkeitsforschung (CEN), Universität Hamburg, Bundesstraße 55, 20146 Hamburg, Germany
[2]Met Office, FitzRoy Road, Exeter, UK, EX1 3PB
[3] Department of Space, Earth and Environment, Chalmers University of Technology, 41296 Gothenburg, Sweden

*Correspondence to:* Manfred Brath (manfred.brath@uni-hamburg.de)

**Abstract.** A neural network (NN)-based retrieval method to determine the snow ice water path (SIWP), liquid water path (LWP), and integrated water vapor (IWV) from millimeter and sub-millimeter brightness temperatures, measured by using airborne radiometers (ISMAR and MARSS), is presented. The NNs were trained by using atmospheric profiles from the ICON numerical weather prediction (NWP) model and by radiative transfer simulations using the Atmospheric Radiative Transfer Simulator (ARTS). The basic performance of the retrieval method was analyzed in terms of offset (bias) and the median fractional error (MFE), and the benefit of using submillimeter channels was studied in comparison to pure microwave retrievals. The retrieval is offset free for $SIWP > 0.01\,kg\,m^{-2}$, $LWP > 0.1\,kg\,m^{-2}$ and $IWV > 3\,kg\,m^{-2}$. The MFE of SIWP decreases from $100\%$ at $SIWP = 0.01\,kg\,m^{-2}$ to $20\%$ at $SIWP = 1\,kg\,m^{-2}$ and the MFE of LWP from $100\%$ at $LWP = 0.05\,kg\,m^{-2}$ to $30\%$ at $LWP = 1\,kg\,m^{-2}$. The MFE of IWV for $IWV > 3\,kg\,m^{-2}$ is $5\%$ to $8\%$. The SIWP retrieval strongly benefits from sub-millimeter channels, which reduce the MFE by a factor of two, compared to pure microwave retrievals. The IWV and the LWP retrieval also benefit from sub-millimeter channels, albeit to a lesser degree. The retrieval was applied to ISMAR and MARSS brightness temperatures from FAAM flight B897 on 18 March 2015 of a precipitating frontal system west of the coast of Iceland. Considering the given uncertainties, the retrieval is in reasonable agreement with the SIWP, LWP, and IWV values simulated by the ICON NWP model for that flight. A comparison of the retrieved IWV with IWV from 12 dropsonde measurements shows an offset of $0.5\,kg\,m^{-2}$ and an rms difference of $0.8\,kg\,m^{-2}$, showing that the retrieval of IWV is highly effective even under cloudy conditions.

## 1  Introduction

Ice clouds are in an ongoing focus of atmospheric remote sensing as they play an important role in atmospheric radiation due to their reflection of sun light and due to their entrapment of infrared radiation. The bulk mass of ice in the atmosphere is typically used to describe the column-integrated bulk mass of atmospheric ice, also known as the ice water path (IWP). Measuring the IWP continues to remain a challenging task and is an important gap in the current global climate observation system. Buehler et al. (2012b) and Holl et al. (2014) argued that this discrepancy is one of the reasons why there are large differences in the

IWP estimates in climate models. In general, the term IWP is defined for the whole integrated ice bulk mass, for example in the work of Evans et al. (2012) and Holl et al. (2014). However, in this paper, henceforth, we distinguish between cloud ice, which consists mainly of ice particles with diameters $< 100\,\mu m$, and snow, which consists mainly of ice particles with diameters $> 100\,\mu m$. This threshold results from the used particle size distribution (see Sect. 3.2). This distinction between small and large ice particles is similar to that in atmospheric models such as the Icosahedral Nonhydrostatic (ICON) model (Zängl et al., 2015) or in the European Centre for Medium-Range Weather Forecasts (ECMWF) IFS-137 model (Eresmaa and McNally, 2014). Hereinafter, we define the CIWP as the column integrated bulk mass of cloud ice and we define the snow water path (SIWP) as the column integrated bulk mass of snow. Note that snow defined in this way can and does occur at high altitudes; typical cirrus clouds in the used model fields contained about 2/3 of their mass in the form of snow, and only the remaining in the form of cloud ice.

Existing methods to estimate the IWP use passive sensors within the microwave, infrared, and visible ranges of the electromagnetic spectrum, and use active sensors such as radar or lidar, or combinations of different sensors. Comprehensive overviews of existing methods can be found in Eliasson et al. (2013) and Holl et al. (2014). According to Holl et al. (2014) active sensors especially combined radar/lidar are probably capable of estimating IWP with a higher accuracy than any existing passive sensor. Furthermore, because of the principle on which their measurements are based, active sensors such as lidar and radar are much more suited for also retrieving the vertical structure. The problem with active sensors is that they lack horizontal coverage, because they only sample the atmosphere directly below the satellite.

Existing passive sensors are problematic in that their sensitivity is highly selective. Passive microwave sensors for example lack sensitivity for thin ice clouds, but are capable of sensing the whole column, whereas infrared and optical sensors are capable of sensing thin ice clouds but cannot sense the whole column because high clouds obscure lower clouds. Sub-millimeter waves are much more sensitive to ice clouds compared to microwaves, as we show in Sect. 3, but passive sub-millimeter waves are still capable of sensing the whole column in contrast to infrared or visible waves. The use of sub-millimeter waves therefore ensures that the retrieval of the IWP based on combined microwave and sub-millimeter wave measurements is more effective than when using infrared or visible waves. This approach also obviates the need for collocating data from different sensors, for example when using the SPARE-ICE product (Holl et al., 2014). However, regardless of the technique that is used, remote sensing of ice clouds is a difficult task because of the many factors that can influence the measurement (Evans et al., 2012).

The launch of the Meteorological Operational Satellite - Second Generation - B (MetOp-SG B) is planned for the early 2020s. Among other sensors, this satellite will be equipped with an Ice Cloud Imager (ICI) and Microwave Imager (MWI). ICI will be the first operational space borne down-looking sensor with the ability to measure in the sub-millimeter range of the electromagnetic spectrum. The main purpose of ICI, as indicated by its name, is to sense ice clouds. Even though the studies of Buehler et al. (2012b), Buehler et al. (2007) and Jiménez et al. (2007) were not explicitly carried out for ICI, they provide a useful overview of the fundamentals of ICI. The International SubMillimetre Airborne Radiometer (ISMAR) is an airborne radiometer that measures at several frequencies between $118\,GHz$ and $664\,GHz$ of the electromagnetic spectrum. One of the main tasks of ISMAR is to serve as a satellite demonstrator for ICI (Fox et al., 2017). Apart from ISMAR, an other airborne radiometer that measures in a similar region of the electromagnetic spectrum is the Compact Scanning Sub-millimeter-wave

Imaging Radiometer (CoSSIR). Evans et al. (2012) used measurements from CoSSIR on board the ER-2 aircraft to estimate the IWP.

In March 2015, COSMICS (Cold-air Outbreak and Sub-Millimetre Ice Cloud Study) was carried out around the northern part of the United Kingdom and Iceland. Among other measurements, COSMICS recorded airborne radiometer measurements with ISMAR and the Microwave Airborne Radiometer Scanning System (MARSS). These measurements were conducted using the BAe-146 aircraft from the Facility for Airborne Atmospheric Measurements (FAAM). ISMAR and MARSS together cover most of the ICI and MWI channels $\geq 89\,\mathrm{GHz}$, which makes ISMAR and MARSS very useful in view of MetOp-SG B.

The main purpose of this work is to develop a method to retrieve the paths of ice and snow in the atmosphere, known as frozen hydrometeors, from data recorded by airborne millimeter/submillimeter radiometer and to apply the retrieval on real observations. Our plan is to base the retrieval method on artificial neural networks. The artificial neural networks are trained using a database of atmospheric profiles taken from a numerical weather prediction model and associated brightness temperatures calculated using a radiative transfer model. The model profiles are broadly representative of the conditions during the flight, but they span a much greater range of atmospheric conditions. As the simulations need information about cloud liquid water, precipitating water, and water vapor, we additionally investigated retrieval for column integrated cloud liquid water, which we term the liquid water path (LWP), the column integrated precipitating water, which we term the rain water path (RWP), and the column integrated bulk mass of water vapor, which we term integrated water vapor (IWV). Our retrieval approach is similar to a previous approach of Jiménez et al. (2007). However, our study differs from theirs in three main respects: first, we apply the retrieval method to real measurements; second, we are not only interested in frozen hydrometeors; and, third, our system can be employed over the ocean, whereas the approach of Jiménez et al. (2007) worked only over land. The performance of the neural network retrieval is evaluated using an independent set of atmospheric profiles and simulated brightness temperatures to get an error estimate of the retrieval. Furthermore, the retrieval is applied to the observation and the retrieved quantities are compared to numerical weather prediction model values as a consistency check . Although Wang et al. (2016) followed a similar approach to estimate hydrometeor paths, they did not apply their approach to measured data. They only used measurement data up to $200\,\mathrm{GHz}$ to validate their simulations. In contrast to our study, the retrieval system they developed was intended for retrieval over land and ocean.

The text is structured as follows: In Sect. 2 we provide an overview of ISMAR and MARSS. In Sect. 3 we describe the retrieval method. This includes the basic assumptions of the method, the structure and training of the artificial neural networks we used, the approach we followed to conduct the simulations, and the approach we used to construct the dataset on which to train the neural network and to check the consistency of the simulations. Sect. 4 contains the results of our test of the retrieval system under ideal conditions to obtain the limits of the procedure and a discussion of the results. In Sect. 5 we present the results of applying the retrieval method to ISMAR and MARSS measurements and discuss the results. In Sect. 6 we summarize the results.

## 2 Sensors

### 2.1 ISMAR

ISMAR is an along-track scanning heterodyne radiometer, which measures between $118\,\text{GHz}$ and $664\,\text{GHz}$ (Table 1). ISMAR is jointly funded by the UK Met Office and the European Space Agency (ESA). One task of ISMAR is to serve as an airborne demonstrator for the upcoming ICI on MetOp-SG B. ISMAR measures at similar frequencies as ICI except for the channels at approximately $118\,\text{GHz}$, which form part of MWI instead, and which is on board the same satellite. ISMAR measures radiation as Rayleigh-Jeans calibrated brightness temperatures. This means, within ISMAR, the received radiation power is converted to brightness temperatures using the Rayleigh-Jeans approximation for a blackbody. Except for the window channels at $243.2\,\text{GHz}$ and $664.0\,\text{GHz}$, ISMAR measures single linear polarization. The window channels measure dual orthogonal linear polarization. ISMAR is mounted on the left side of the aircraft allowing both upward and downward views. Downward views with nominal nadir incidence angles between +50 and -10 degrees are possible, where positive angles indicate directions towards the front of the aircraft. Zenith observations can be made in the +10 to -40 degree range. The nadir +50 degree view is designed to give a close match in incidence angle to conically-scanning imagers such as ICI. However, in this work we use only the near-vertical nadir view in order to eliminate any polarization differences. For further details on ISMAR see Fox et al. (2017). Polarization differences are not expected in the vertical view as both polarizations are orthogonal to both the surface and the clouds, and the sensed medium is likely to be random in the azimuth direction. Therefore, the two polarizations of the window channels were averaged.

### 2.2 MARSS

MARSS is an along-track scanning heterodyne radiometer, which measures between $89\,\text{GHz}$ and $183\,\text{GHz}$ (Table 1). The viewing directions of MARSS are $40°$ to $-40°$ nadir and $40°$ to $-40°$ zenith. MARSS is an airborne version AMSU-B (McGrath and Hewison, 2001). MARSS is also mounted on the side of the aircraft allowing similar upward and downward views. MARSS measures single linear polarization and measures the radiation as Rayleigh- Jeans calibrated brightness temperatures. Further details on MARSS can be found in McGrath and Hewison (2001) and the articles cited therein. In this work, we use only the nadir viewing direction.

## 3 Retrieval method

Retrieving hydrometeor paths from brightness temperatures or in general from the radiance is an inverse problem with the generic form (Rodgers, 2000):

$$\boldsymbol{Y} = f\left(\boldsymbol{X}\right) + \varepsilon \tag{1}$$

where $\boldsymbol{Y}$ is the vector of the measured brightness temperatures, $\boldsymbol{X}$ is the vector quantities to retrieve, $f$ (the forward model) is the radiative transfer and sensor model, that can simulate brightness temperatures for a given atmospheric state, and $\varepsilon$ is the

**Table 1.** Channel description, taken mostly from Fox et al. (2017).

| # | Center fre-quency [GHz] | Side bands [GHz] | Band-widths [GHz] | Noise [K] | Polari-zation | Instrument | Feature |
|---|---|---|---|---|---|---|---|
| 1 | 89.0 | ±1.1 | 0.65 | 0.5 | v | MARSS | window |
| 2 | 118.75 | ±1.1 | 0.4 | 0.2 | v | ISMAR | oxygen line |
| 3 | 118.75 | ±1.5 | 0.4 | 0.2 | v | ISMAR | oxygen line |
| 4 | 118.75 | ±2.1 | 0.8 | 0.2 | v | ISMAR | oxygen line |
| 5 | 118.75 | ±3.0 | 1.0 | 0.2 | v | ISMAR | oxygen line |
| 6 | 118.75 | ±5.0 | 2.0 | 0.2 | v | ISMAR | oxygen line |
| 7 | 157.05 | ±2.6 | 2.6 | 0.5 | v | MARSS | window |
| 8 | 183.31 | ±1.0 | 0.45 | 0.5 | v | MARSS | water vapor line |
| 9 | 183.31 | ±3.0 | 1.0 | 0.5 | v | MARSS | water vapor line |
| 10 | 183.31 | ±7.0 | 2.0 | 0.5 | v | MARSS | water vapor line |
| 11 | 243.20 | ±2.5 | 3.0 | 0.3, 0.5 | h, v | ISMAR | window |
| 12 | 325.15 | ±1.5 | 1.6 | 1.1 | v | ISMAR | water vapor line |
| 13 | 325.15 | ±3.5 | 2.4 | 0.3 | v | ISMAR | water vapor line |
| 14 | 325.15 | ±9.5 | 3.0 | 0.8 | v | ISMAR | water vapor line |
| 15 | 448.0 | ±1.4 | 1.2 | 0.9 | v | ISMAR | water vapor line |
| 16 | 448.0 | ±3.0 | 2.0 | 1.3 | v | ISMAR | water vapor line |
| 17 | 448.0 | ±7.2 | 3.0 | 1.9 | v | ISMAR | water vapor line |
| 18 | 664.0 | ±4.2 | 3.0 | 0.9, 2.7 | h, v | ISMAR | window |

measurement noise. The typical inverse problem in remote sensing is an ill-proposed problem. Many different ways have been reported in the literature to overcome this problem, for example optimal estimation (Rodgers and Connor, 2003), Monte Carlo integration in combination with Bayesian inference (Evans et al., 2012), or artificial neural networks (NN) (Defer et al., 2008; Jiménez et al., 2007). We followed the latter approach and used neural networks to retrieve the desired quantities. For a detailed introduction on neural networks, see for example Rojas (2013). Before it can be used, a neural network requires training data to set up the network. Construction of the training dataset is explained in the next subsection. Details of the neural network follow in Sect. 3.4.

## 3.1 Training Database

The training database plays a crucial role in neural network-based retrieval. All the assumptions on which the retrieval method is based are condensed in the database. For example, the database needs to cover the actual measurement space (the full range of $Y$'s), failing which the retrieval would be unsuccessful for some measurements. This would imply that the assumptions about the atmosphere and the interaction with electromagnetic radiation were inadequate. Therefore, it is important to make reasonable assumptions. The two main assumptions in terms of retrieval are that the atmospheric profiles from a numerical

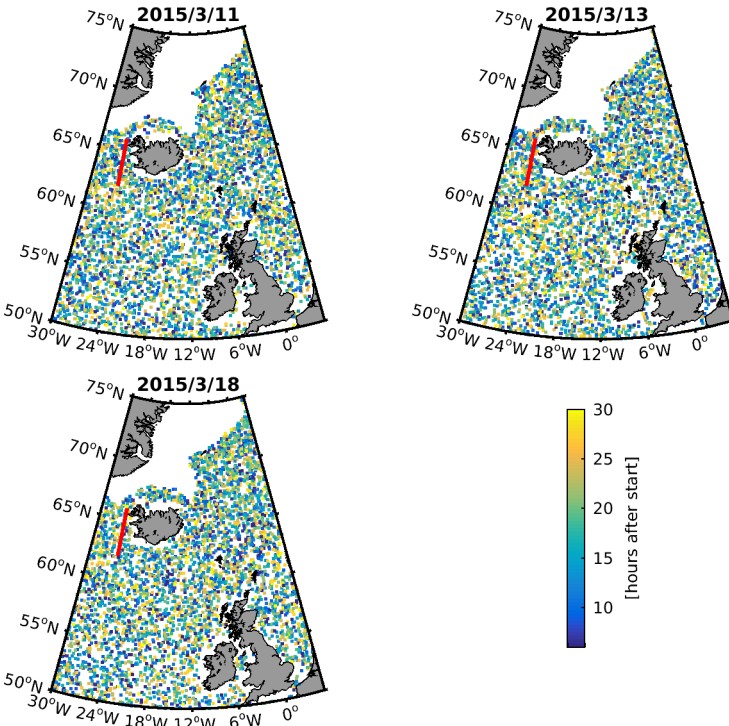

**Figure 1.** Locations of the approximately 13,000 randomly selected profiles of the training database from the three ICON forecast runs on March 11, 13, and 18, 2015. Each dot stands for one profile. The day time of each profile is color indicated in hours after start of the runs. The red line depict the north-south-transects of the FAAM flight B897 on March 18, 2015. The white ocean areas north of Iceland and east of Greenland indicate areas covered with sea ice.

weather prediction (NWP) model are sufficient to describe the possible states of the atmosphere and that the interaction of the atmosphere with the electromagnetic radiation can be described by a radiative transfer model.

We use atmospheric profiles from simulations of a regional version of the ICON model, details of which can be found in Zängl et al. (2015) and Reinert et al. (2016). The atmospheric profiles were taken from three ICON forecast runs on March 11, 13, and 18, 2015 of the region between 50°N and 75°N and 30°W and 5°E with a gridded resolution of about 10 km. These three runs cover the three different FAAM BAe-146 flights during COSMICS. Each run started at 0h00 GMT using the starting conditions from the Integrated Forecasting System (IFS) of the European Centre for Medium-Range Weather Forecasts (ECMWF) and ended at 6h00 GMT on the following day. The complete model fields of each run had a time resolution of 30 minutes. The fields for the first six hours of each run were excluded to eliminate possible spin-up effects. From the remaining time steps we randomly selected approximately 13,000 atmospheric profiles over the ocean. The location and the time of the profiles and the north-south-transects of the FAAM flight B897 on March 18 are shown in Fig. 1. The selected profiles cover a much wider range of atmospheric conditions than the actual conditions during the flight. The flight took about 3 hours

west of Iceland, whereas the selected profiles span in total a time range of 72 hours over a much larger area than the actual flight. Because the atmospheric profiles were from the same season and they cover a wide range of atmospheric conditions including the conditions during the flight, these profiles are expected to sufficiently cover the situations encountered during the measurement flight without being optimized for this specific flight. Though, the database covers a wide range of atmospheric

conditions, it is constrained to similar season and similar latitude range over ocean. A retrieval based on this database is likely to provide insufficient results, when applied to different season, to different latitute or even over land. Simulated brightness temperature measurements for each atmospheric profile were generated for the database using the Atmospheric Radiative Transfer Simulator (ARTS) (Eriksson et al. (2011) and Buehler et al. (2005)).

## 3.2 Radiative Transfer Simulations

ARTS, which is a radiative transfer model for thermal radiation, can process fully polarized radiative transfer calculations with scattering. This is important as microwave and sub-millimeter radiation mostly interacts with ice particles by scattering. We used ARTS version 2.3. The Discrete Ordinate ITerative (DOIT; Emde, 2004) method was used as scattering solver within ARTS. The Rayleigh-Jeans brightness temperatures were simulated for each randomly selected atmospheric profile. No explicit spectral response function was used to simulate the ISMAR and MARSS channels; instead, we conducted monochromatic

radiative transfer simulations for the center frequencies of the two side bands of each channel and obtained their average. Tests with highly spectral resolved clearsky simulations showed that the error by using only the center frequency of each pass band is $< 1\,\mathrm{K}$. Possible effects due to different footprint sizes and beam-filling are neglected as the footprints of MARSS and ISMAR are much smaller than similar satellite instruments, or an ICON model grid cell. The footprint size at ground level is pretty much the same for all the ISMAR channels and are in the order of $700\,\mathrm{m}$ for a flight altitude of $10\,\mathrm{km}$. The footprint size at the

surface varies with channel; for a flight altitude of 10km it varies between 700 and 1500m.

Within ARTS, gas absorption was taken into account by using the HITRAN database (Rothman et al., 2013) and the MT_CKD model for the continuum absorption of water vapor and molecular nitrogen in version 2.52 (Mlawer et al., 2012). The gas absorption of molecular oxygen was processed by using the full absorption model of Rosenkranz (1998) modified by the values from Tretyakov et al. (2005). The surface emissivity was calculated using the FAST microwave Emissivity Model

(FASTEM; Liu et al., 2011) implementation within ARTS 2.3 using the surface wind speed and surface temperature from the ICON model dataset. Although FASTEM was originally developed for low microwave frequencies, with further development the valid frequency range was enhanced to higher frequencies. Liu et al. (2011) tested FASTEM up to $150\,\mathrm{GHz}$. Prigent et al. (2016) compared FASTEM with the Tool to Estimate Sea-Surface Emissivity from Microwaves to sub-Millimeter waves (TESSEM2) and with ISMAR measurements from two low-level flights at low wind speeds. They showed that FASTEM tends

to underestimate the emissivity at $243.3\,\mathrm{GHz}$ leading to errors of order $5\,\mathrm{K}$ in the upwelling brightness temperature close to the surface (flight altitudes $< 300\,\mathrm{m}$). The emissivity using FASTEM at $243\,\mathrm{GHz}$ is roughly between $0.7$ and $0.8$ for nadir viewing direction and atmospheric conditions during FAAM flight B897. At surface level and for a surface temperature of $276\,\mathrm{K}$, which is the surface temperature in the ICON model for the beginning of the first transect of FAAM flight B897 (see also Sect. 5), these emissivities result in upwelling brightness temperatures of $193\,\mathrm{K}$ and $221\,\mathrm{K}$ and a difference of 28K in the upwelling

brightness temperatures. Clearsky simulations using ARTS for conditions similar to the driest conditions during the FAAM flight B897 show for an IWV of $6\,\mathrm{kg\,m^{-2}}$ at a flight level of $10\,\mathrm{km}$ an upwelling brightness temperature of $233\,\mathrm{K}$ for a surface emissivitiy of 0.7 and $243\,\mathrm{K}$ for a surface emissivitiy of 0.8. The difference in upwelling brightness temperatures is reduced to $10\,\mathrm{K}$ at a flight level of $10\,\mathrm{km}$. This is roughly one third of the upwelling brightness temperature difference at surface level.

So, a $5\,\mathrm{K}$ error in the upwelling brightness temperature at the surface will result in a worst-case error of approximately $1.8\,\mathrm{K}$ at $10\,\mathrm{km}$. For greater IWV the error is even smaller. Therefore, considering the strong scattering signal at $243.3\,\mathrm{GHz}$ (see Fig. 3), we do not consider this problematic. For the higher frequency ISMAR channels ($325\,\mathrm{GHz}$ and higher, Ch. 12-18) the effect of surface emissivity errors will be smaller due to the strong water vapour absorption at these frequencies.

Each atmospheric profile consists of the following profiles with 90 pressure levels between $0.02\,\mathrm{hPa}$ and the surface pressure:

Atmospheric temperature in K, altitude in m, atmospheric humidity in vmr, cloud liquid water in $\mathrm{kg\,m^{-3}}$, cloud ice water in $\mathrm{kg\,m^{-3}}$, rain in $\mathrm{kg\,m^{-2}s^{-1}}$, and snow in $\mathrm{kg\,m^{-3}}$. Oxygen and nitrogen levels were assumed to be constant with volume mixing ratios of 0.2095 and 0.7808, respectively.

The ICON runs used a 1-moment microphysics scheme with four distinct hydrometeor types namely liquid cloud water, cloud ice, rain, and snow. Assumptions on particle size distributions and shape are necessary in order to simulate brightness

temperatures. Our assumptions are similar to Geer and Baordo (2014) with one exception: Geer and Baordo (2014) use sector-like snowflakes from the Liu (2008) -database to simulate the scattering of snow. The Liu database is valid only for frequencies up to $340\,\mathrm{GHz}$, which is insufficient for our simulations. Instead, we use aggregates from the database of Hong et al. (2009) to simulate the scattering of snow, because the Hong aggregate is the only aggregate habit for which there exists publicly available data above $340\,\mathrm{GHz}$. According to Eriksson et al. (2015), Hong aggregates reasonably represent the average scattering

properties of snow. However, in some respects the Hong database is also problematic. Firstly, the effective density of the Hong aggregates is constant, whereas the effective density of snow changes with the particle size. Secondly, the data are based on the old Warren (1984) refractive index data, which do not include the temperature dependencies. We therefore used a corrected version of the Hong et al. (2009)-database in which the absorption is rescaled using the Mätzler (2006) parametrization for the refractive index of ice. Rescaling is achieved by multiplication with $\mathrm{imag}(n)/\mathrm{imag}(n_0)$, where $n_0$ and $n$ is the refractive

index from Warren (1984) and Mätzler (2006), respectively. The rescaling is used to obtain data for 183, 213, 243 and $266\,\mathrm{K}$. The scattering extinction and all six of the phase matrix values are maintained constant. This means that only the absorption is rescaled. Our assumptions about the microphysics are the same in terms of the basic hydrometeor types but differ from the internal microphysics of the ICON model in terms of size, shape, and density. However, this is not considered an issue, because the function of the ICON model for the database is simply to deliver physically realistically profiles, which span the range of

conditions that may be encountered. For this case, it is not needed to be fully consistent with the ICON model. If the interest is in the ICON microphysics then consistency would be needed.

Explicitly, we used the following four hydrometeors for the radiative transfer simulations:

**Table 2.** Parameters used for the modified gamma distribution

| Hydrometeor | $\mu$ | $\gamma$ | $\Lambda$ |
|---|---|---|---|
| Cloud liquid water | 2 | 1 | $2.13 \times 10^5$ |
| Cloud ice water | 2 | 1 | $2.05 \times 10^5$ |

1. Liquid cloud water: The scattering properties were calculated under the assumption of a spherical shape using the Mie theory. The size distribution was calculated using a modified gamma distribution

$$N(D) = N_0 D^\mu \exp(-\Lambda D^\gamma) \tag{2}$$

   where $D$ the diameter of the spheres using the coefficients of Geer and Baordo (2014). The parameters $\mu$, $\gamma$, and $\Lambda$ are provided in Table 2. The scale parameter $N_0$ is set according to the mass concentration using the expression for the third moment of a modified gamma distribution (Petty and Huang, 2011)

2. Cloud ice: The scattering properties were calculated under the assumption of a soft sphere with a density of $900 \, \text{kg/m}^3$ using the Mie theory as in Geer and Baordo (2014). The size distribution was calculated using a modified gamma distribution (Eq. 2). The parameters $\mu$, $\gamma$, and $\Lambda$ are listed in Tab. 2. The scale parameter $N_0$ is set according to the mass concentration using the expression for the third moment of a modified gamma distribution (Petty and Huang, 2011).

3. Rain: The scattering properties were calculated under the assumption of a spherical shape using the Mie theory. The size distribution was calculated using the Marshall-Palmer size distribution (Marshall and Palmer, 1948), for which the mass flux was converted to the rain rate by assuming a constant density of $1,000 \, \text{kgm}^{-3}$.

4. Snow: We assume snowflakes behave similar to the aggregates from the Hong- DDA database (Hong et al., 2009). The size distribution was calculated using the midlatitude version of the distribution from Field et al. (2007). The mass-dimension relationship we used is

$$m(D) = \alpha \left( \frac{D}{D_0} \right)^\beta \tag{3}$$

   where $\alpha = 65.4 \, \text{kg}$ and $\beta = 3$ are the shape parameters, $D$ is the maximum diameter, and $D_0$ is the unit maximum diameter. The shape parameters $\alpha$ and $\beta$ were calculated from the shape dimensions.

The selected size distributions define the size range covered by the different hydrometeor habits. These choices result in cloud ice mainly consisting of particles $< 100 \, \mu\text{m}$, whereas snow mainly consist of particles $> 100 \, \mu\text{m}$.

### 3.3 Comparison of Simulations and Measurements

Before we can start with the retrieval, we have to verify whether the data in our training database covers the measurements. If the simulations do not cover the full range of measurements or only partially cover this range, the retrieval is likely to provide

insufficient results. In Fig. 2 the brightness temperature of each channel at a flight altitude of $10,500\,\mathrm{m}$ is plotted against that of all the other channels, such that the plot consists of 18 times 18 subplots. The diagonal is empty by definition. The channels stated above the plots correspond to the brightness temperatures on the X-axis and the channels stated on the right-hand side correspond to the brightness temperatures on the Y-axis. The plot in Fig. 2 shows how each channel is correlated with every other channel. First, let us consider the upper right half of the plot, where the measurements are plotted over the simulations. Although the measured values cover a smaller area than the simulated values, the former of these values are mostly surrounded by the latter values. This means that the variability of our simulations is higher than the variability of the measurements. As we chose the profiles randomly we do not expect to obtain an exact match between each measurement and its simulation. Actually, this is not necessary and is not our intention. The ICON profiles only have to be physically realistic and span the possible range of conditions. The important point is that the set of measurements is contained within the set of simulations. In the lower left half, where the simulations are plotted over the measurements, we can easily determine whether the set of measurements is within the set of simulations. Mostly, the red dots are covered by the blue dots, meaning the measurements are within the set of simulated values. The simulated brightness temperatures of the $183.31 \pm 1\,\mathrm{GHz}$ channel, the $325.15 \pm 1.5\,\mathrm{GHz}$ channel, the three $448\,\mathrm{GHz}$ channels, and the $664\,\mathrm{GHz}$ channel are slightly higher than the measured brightness temperatures. One reason could be the presence of an insufficient amount of water vapor in the upper troposphere of the randomly selected atmospheric profiles from the ICON model dataset, because these channels are sensitive to the upper troposphere. Another reason could be the spectroscopy used within ARTS.

Of course this comparison cannot prove the sufficiency of our training database for retrieval purposes; however, thus far the behavior seems to be reasonable and understandable. We therefore expect the training database to be adequate for the retrieval.

Before we discuss the neural network, we investigate the influence of frozen hydrometeors on the brightness temperatures. The liquid particles interact with the electromagnetic radiation by absorption and in the case of rain also by scattering, whereas the frozen particles interact with the electromagnetic radiation mainly by scattering. Furthermore, the absorption is mostly related to the total mass of the particles and is less dependent on the particle size, whereas scattering strongly depends on the particle size. Buehler et al. (2007) showed, for frequencies similar to those of ISMAR, that the frozen particles must have an effective diameter $> 100\,\mathrm{\mu m}$ to have a significant influence on the brightness temperatures. Fig. 3 (a) shows the difference in brightness temperatures between a subset of 450 simulations without cloud ice and with cloud ice as a function of the CIWP. The maximum difference is $< 0.5\,\mathrm{K}$, which is mostly smaller than the noise of ISMAR and MARSS. This means, by using ISMAR and MARSS, there is no possibility to physically sense CIWP, bearing in mind, that within this study, CIWP is the column integrated bulk mass of ice particles mostly smaller than $100\,\mathrm{\mu m}$. In this respect, our work contrasts with that of Wang et al. (2016), who stated that they can estimate CIWP. The reason for this difference is that they assume a different particle size distribution for cloud ice, which results in larger cloud ice particles. Fig. 3 (b) shows the difference in brightness temperatures between a subset of 450 simulations without snow and with snow as a function of the SIWP. A clear relationship between the SIWP and the difference in brightness temperature can be seen. The difference in brightness temperature is up to $50\,\mathrm{K}$. For the $243\,\mathrm{GHz}$ channel, it is even up to $80\,\mathrm{K}$ (outside the y-axis range of Figure 3).

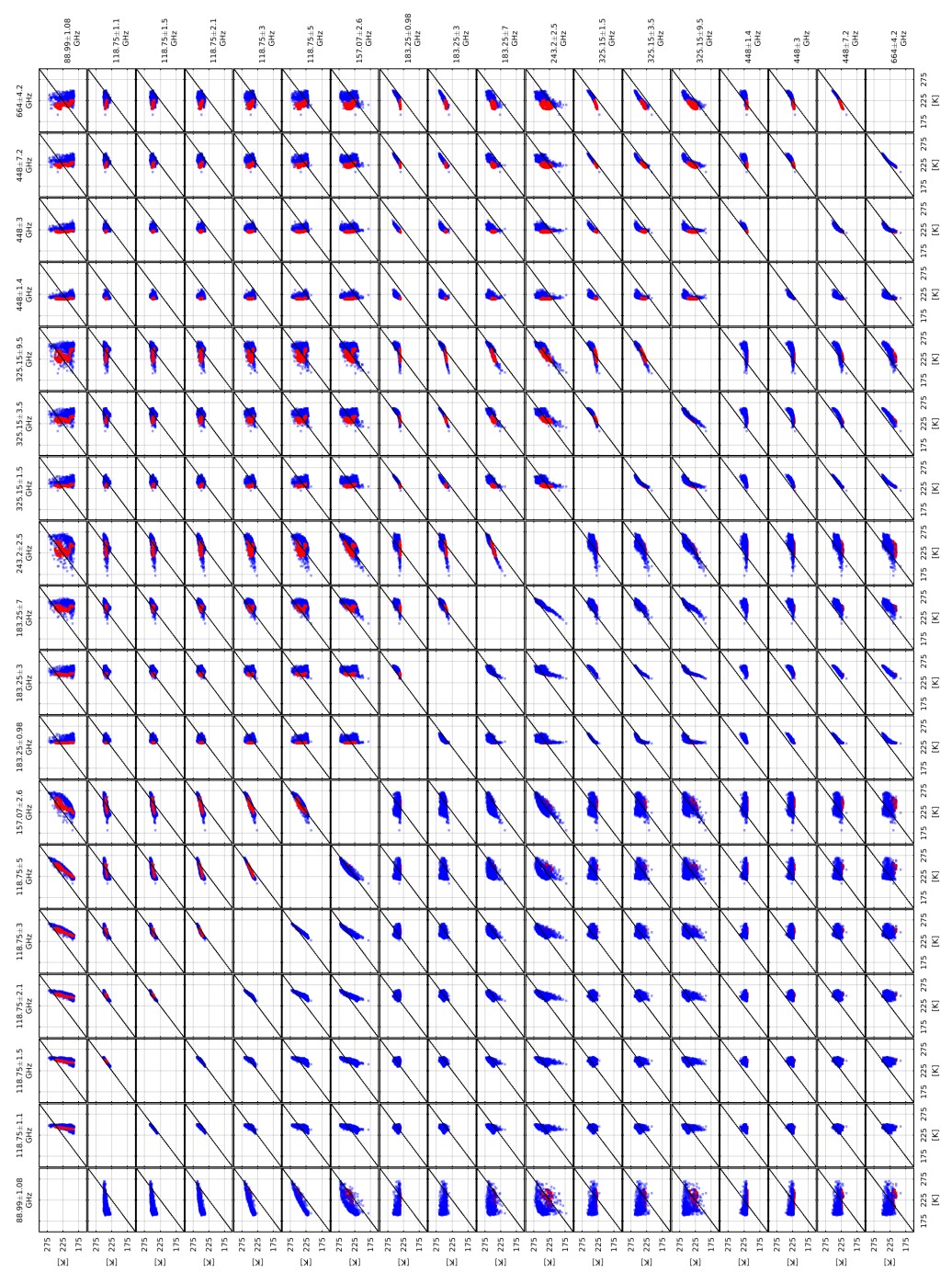

**Figure 2.** Comparison of simulated brightness temperature (blue) and measured brightness temperature of flight B897 (red). The channels indicated above and on the right correspond to the brightness temperatures on the X- and Y-axes, respectively.

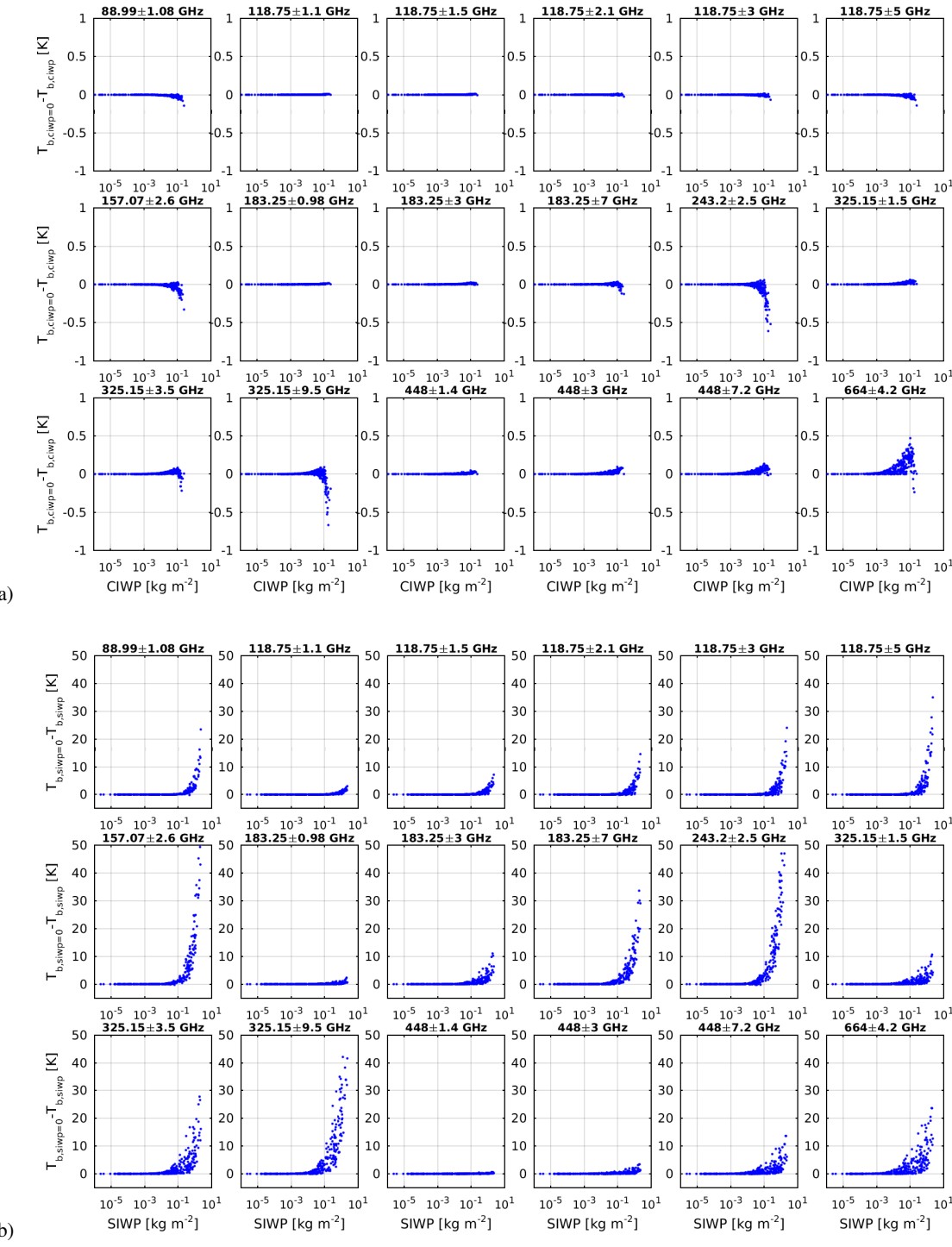

**Figure 3.** Difference in brightness temperatures between simulations with (a) cloud ice set to zero and with simulation with unchanged cloud ice as a function of the CIWP, (b) same as for (a) but with snow set to zero and as a function of SIWP.

### 3.4 Neural network

Before the neural network is set up, the retrieval method has to be defined. The main interest of this study is to retrieve SIWP, but also to investigate the retrieval of IWV, LWP and RWP. Except for IWV, these quantities have a high dynamic range and all four quantities are always greater than or equal to $0\,\mathrm{kg\,m^{-2}}$. Therefore, we retrieve the logarithm of the ratio of the desired quantity and the unit path, for example for SIWP:

$$siwp = \log_{10} \frac{SIWP}{SIWP_0} \qquad (4)$$

with $SIWP_0 = 1\,\mathrm{kg\,m^{-2}}$ the unit snow water path. As the logarithm is not defined for zero, every zero value of the four quantities is assigned the value of $10^{-9}\,\mathrm{kg\,m^{-2}}$ before computing the logarithm, which was the order of the smallest values above zero, to avoid infinite values. Thus, the smallest value of a retrieval quantity is $-9$. Henceforth, writing the SIWP or one of the other three quantities in lowercase means that the decadic logarithm of the quantity was used. Our state vector $\boldsymbol{X}$ refers to

$$\boldsymbol{X} = \begin{pmatrix} lwp \\ rwp \\ siwp \\ iwv \end{pmatrix}. \qquad (5)$$

The measurement vector $\boldsymbol{Y}$ consists of $18$ components. Each component is the measured brightness temperature $T_b$ of one of the 18 combined channels of ISMAR and MARSS

$$\boldsymbol{Y} = \begin{pmatrix} T_{b,ch1} \\ T_{b,ch2} \\ \vdots \\ T_{b,ch18} \end{pmatrix} \qquad (6)$$

with the channels defined as in Table 1.

Instead of using one neural network for the retrieval, we use an ensemble of neural networks. According to Heskes et al. (1997), an ensemble of neural networks is expected to provide a more accurate estimate of the true regression than would be possible with only one neural network. The retrieved state vector $\boldsymbol{X}$ is then the average over the estimated state vectors of each neural network of the ensemble,

$$\boldsymbol{X} = \frac{1}{N} \sum_{n=1}^{N} \boldsymbol{X}_n, \qquad (7)$$

where $N$ is the number of neural networks and $\boldsymbol{X}_n$ is the estimated state vector of the $n$-th neural network. An ensemble of 20 neural networks is used for the retrieval. Each neural network consists of one input, one hidden, and one output layer with

18, 12, and 4 neurons, respectively. The input neurons are the components of the measurement vector $Y$, i.e., the measured brightness temperatures. The output neurons are the components of the state vector $X$, i.e., the logarithms of the path of the 3 hydrometeors and the logarithm of the IWV. Each neural network is trained with simulated measurement vectors from the training database and the corresponding state vectors. The noise behavior of the measured brightness temperatures is included by adding a Gaussian distributed error to every simulated brightness temperature with a standard deviation of the noise of each channel, see also Table 1.

The neural networks are trained by using approximately $6,000$ state vectors. The other $7,000$ state vectors are used for testing. Each neural network of the ensemble is trained with a randomly chosen subsample of about $3,000$ state vectors and the corresponding measurement vectors. Each neural network is trained by the Levenberg-Marquardt algorithm (Hagan and Menhaj, 1994).

As simple to use and as powerful as neural networks are, these networks have a downside. As soon as one part of the measurement setup is changed, a new neural network must be trained. If the number of channels or even simply the position of one channel is changed, it is necessary to train a new neural network. This has the implication that for airborne measurements, different neural networks are required for different flight altitudes. Nonetheless the computational burden is not high. Once the neural networks are trained, which take some hours, they are very fast. For satellites such as MetOp-SG B, which will carry MWI and ICI, this is less of an issue, because observation will always be from above the top of the atmosphere. The main issue for a satellite application is that the training database must cover the global range of atmospheric conditions. Therefore our retrieval is limited to similar seasons and latitude range as the used database, but there is no fundamental limit in the usage a neural network for global retrieval application, as long the database covers the wide range of globally possible atmospheric conditions and the neural network can capture this variability. For example, Holl et al. (2014) applied their trained neural network globally to retrieve IWP. By using for example similar ICON model runs for several globally distributed regions and different seasons, our retrieval can be expanded to global applications.

## 4 Basic Retrieval Performance

Retrieval simulations for a flight altitude of $10.5\,\mathrm{km}$ are used to test the basic retrieval performance. We applied the neural network, which was trained with one part of the training database, to the other part of the training database. This means, the retrieval procedure was applied to approximately $7,000$ measurement vectors with simulated brightness temperatures. For each of these $7,000$ measurement vectors the corresponding state vector is known. Thus, the results of the retrieval can be compared directly with the true state vectors. This is a test under ideal conditions as retrieval and test data are based on the same assumptions. Possible errors due to radiative transfer simulation or errors of the model profiles are excluded in this test. In that case, the retrieval performance is limited by the errors of the artificial neural networks and from the radiometer noise of MARSS and ISMAR in combination with limited interaction between the electromagnetic radiation and the atmosphere. We excluded the error of the radiative transfer simulation and the error of the atmospheric model, because the modeling errors are difficult to estimate, as there is no data to compare with. Therefore, the errors from the direct comparison are an estimate of the

physical limits of our retrieval approach. The retrieval error when applying the retrieval on measured brightness temperatures is likely to be larger, as the a priori assumptions will be never completely fulfilled.

## 4.1 Offset

In Fig. 4 the difference between the retrieved state vector and the true state vector is shown as a two-dimensional histogram.
The x- and y-axes show the component of the retrieved state vector and the corresponding component of the difference between the retrieved and the true state vector, respectively. On the x- and y-axes, 45 equally sized bins between $-9$ and 2, and 121 bins between $-5$ and 12 are used, respectively. Because of the different value range of IWV, 121 equally sized bins between $-1$ and 2, and 161 bins between $-1$ and 1 are used on the x- and y-axes, respectively. The histograms are normalized with respect to the number of state vectors.

The relative frequency of occurrence is coded as different grey shadings. Recall that the components of the state vectors are logarithmic quantities, as mentioned in the beginning of Sect. 3. The difference in the logarithmic quantities is the same as the logarithm of the ratio of the linear quantities. For example, a y-axis value of 1 in Fig. 4 corresponds to a factor 10 error, and a value of 0.1 corresponds to a 25% relative error. To look for systematic errors of each component, the offset $O_i$ as a function of the $j$-th bin of the binned $i$-th component of the retrieved state vector is shown as a blue line. The offset $O_i$ is

$$O_i(x_{ret,ij}) = \frac{\sum_{k=1}^{161} w_{ijk} \Delta x_{ijk}}{\sum_{k=1}^{161} w_{ijk}}, \tag{8}$$

where $w_{ijk}$ is the number of occurrences of bin $(k, j)$ of component $i$, and $\Delta x_{ijk}$ the binned difference $\Delta x_i = x_{ret,i} - x_{true,i}$ between the component $i$ of the retrieved and the true state vector of bin $(k, j)$. The standard deviation $\sigma_{O,i}$ is calculated to consider the random error. The standard deviation $\sigma_{O,i}$ is shown by red dashed lines on either side of the offset $O_i$. The standard deviation $\sigma_{O,i}$ is

$$\sigma_{O,i}(x_{ret,ij}) = \left[ \frac{\sum_{k=1}^{161} w_{ijk} \left( \Delta x_{ijk} - O_i(x_{ret,ij}) \right)^2}{\sum_{k=1}^{161} w_{ijk}} \right]^{\frac{1}{2}}. \tag{9}$$

The offset and the standard deviation were calculated for each $j$-th binned component of the estimated state vector but only if the summed number of occurrences in the $j$-th bin is at least 1% of the number of state vectors to avoid statistical fluctuation due to small numbers. Strikingly, there is a straight line in the upper half of the plots of the retrieved hydrometeors indicating a bimodal distribution for small values. For values $> -3$ this second mode vanish. These lines depict cases of overestimation
of the specific hydrometeor. All cases on these lines are cases where we set the specific component of the state vector to $-9$ to avoid infinite values, because for these cases the actual hydrometeor path was zero.

The SIWP-histogram has a bell-mouthed shape, from which we can infer that with increasing amount the error decreases. The offset of the retrieved SIWP is 0 for SIWP $> -2$. In addition to this, the standard deviation is symmetric around zero for SIWP $> -2$. For SIWP $< -2$, the offset is oscillating around zero with increasing amplitude for decreasing SIWP. Up to
this point, we can record that for SIWP $> -2$ the retrieval has no offset and the standard deviation decreases from 0.6 to 0.2 with increasing SIWP. The standard deviation of SIWP $> -2$ is of the same order of magnitude as the error for the retrieved

IWP within the work of Evans et al. (2012). These authors used combined passive microwave and sub-millimeter radiometers to retrieve the IWP among other quantities. The IWP of Evans et al. (2012) corresponds to the column integrated bulk mass of atmospheric ice, whereas SIWP is the column integrated bulk mass of snow. However, as the column integrated bulk mass of cloud ice, which is our definition for CIWP, is typically an order of magnitude smaller, the IWP of Evans et al. (2012) corresponds mostly with the SIWP in our retrieval. A detailed comparison with the work of Evans et al. (2012) is difficult since there is no distinct information about the error as function of the IWP as there is in the work of Holl et al. (2014), for example.

The LWP histogram differs from the SIWP histogram. For LWP $< -1$ the LWP histogram consists mainly of a straight line in the upper half and a wider strip in the lower half. The various values in the lower half mean that many estimated values are underestimated. Due to the fact that cases with no LWP are strongly overestimated, the offset has some stronger jumps around zero. For LWP $> -2$ the offset and the standard deviation become smoother with increasing LWP. For LWP $> -1$ the offset changes only slightly with increasing LWP and the standard deviation decreases from $0.8$ to $0.4$ at LWP $> 0$. The RWP histogram is similar to the LWP histogram for RWP $< -1$. For RWP $> -1$, the size of the standard deviation is still similar to the standard deviation of LWP but compared to LWP there is a strong change of the RWP offset with increasing RWP indicating a significant non-zero offset.

The IWV histogram differs strongly from the histograms of the three estimated hydrometeor paths. It has a rectangular shape and the differences are at least one order of magnitude smaller. Except for IWV $> 1.3$ (IWV $> 20\,\mathrm{kg\,m^{-2}}$), the offset over the whole range of values is practically zero and the standard deviation is almost constant with a value of $0.04$. This means the IWV retrieval is offset-free over that range of values. For IWV $> 1.3$, there is a small offset of $0.02$.

In summary, the retrieval is practically offset-free for IWV, for SIWP $> -2$ (SIWP $> 0.01\,\mathrm{kg\,m^{-2}}$) and for LWP $> -1$ (LWP $> 0.1\,\mathrm{kg\,m^{-2}}$). For RWP, this does not hold.

## 4.2 Median Fractional Error (MFE)

Thus far, we know which quantities can be measured offset-free. We next address the retrieval error, which is described using the median fractional error (MFE), which was also used by Holl et al. (2014) to estimate the error of IWP of the SPARE-ICE product. The MFE is defined as follows,

$$MFE = \mathrm{median}\left[\overbrace{\exp_{10}\left(|x_{i,ret} - x_{i,true}|\right) - 1}^{=FE}\right] \tag{10}$$

with $x_i$ the $i$-th component of the estimated state vector and $x_{i,true}$ the $i$-th component of the true state vector. For example,

$$\begin{aligned} MFE_{SWP} &= \mathrm{median}\left[\exp_{10}\left(|swp_{ret} - siwp_{true}|\right) - 1\right] \\ &= \mathrm{median}\left[\exp_{10}\left(\left|\log_{10}\frac{SIWP}{SIWP_{true}}\right|\right) - 1\right]. \end{aligned} \tag{11}$$

For example $100\,\%$ MFE on SIWP means that for half of the considered cases the retrieved value is within the interval $[SIWP_{true}/2, 2 \cdot SIWP_{true}]$. For MFE $< 30\,\%$ it is approximately equal to the relative error. The MFE for each component of the state vector as function of the corresponding component of the estimated state vector is shown as blue lines in

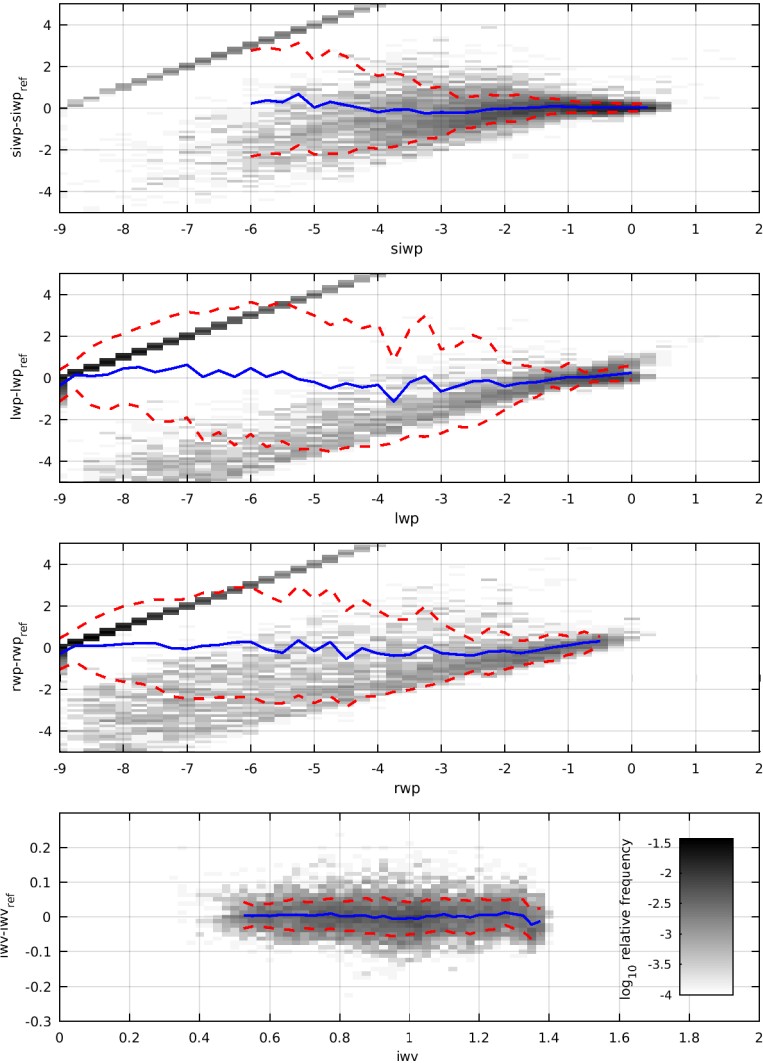

**Figure 4.** Two-dimensional histograms of the difference between components of the retrieved state vector and the corresponding ICON model state vector, as a function of the retrieved state vector. The components of the state vectors are defined as the decadal logarithm of the ratio of the specific quantity and the specific unit path, therefore they are unitless. The offsets $O$ of the components of the retrieved state vector are shown as blue lines and the corresponding standard deviations $\sigma_O$ are shown as red lines. For the hydrometeor paths, an x-value of $-3$ corresponds to a value of $10^{-3}\,\mathrm{kg\,m^{-2}} = 1\,\mathrm{g\,m^{-2}}$. For IWV, an x-value of 1 corresponds to a value of $10^{1}\,\mathrm{kg\,m^{-2}} = 10\,\mathrm{kg\,m^{-2}}$.

Fig. 5. To compute the MFE, the components of the state vector were binned on a logarithmic grid with $45$ bins starting from $10^{-9}\,\mathrm{kg\,m^{-2}}$ and ending at $10^2\,\mathrm{kg\,m^{-2}}$. The different value range for the MFE of IWV necessitated the use of a logarithmic grid with $121$ bins starting from $10^{-1}\,\mathrm{kg\,m^{-2}}$ and ending at $10^2\,\mathrm{kg\,m^{-2}}$ was used for IWV. The MFE is shown only for bins that include at least $1\%$ of the total number of state vectors to avoid statistical fluctuations.

The MFE of SIWP decreases with increasing SIWP. Whereas the MFE of SIWP is more than $600\%$ for SIWP $< 10^{-3}\,\mathrm{kg\,m^{-2}}$ it decreases to $20\%$ at $SIWP = 1\,\mathrm{kg\,m^{-2}}$. For SIWP $> 0.01\,\mathrm{kg\,m^{-2}}$ the MFE is less than $100\%$ and for SIWP $> 0.1\,\mathrm{kg\,m^{-2}}$ the MFE is less than $50\%$, which is in good agreement with the relative error of SIWP over the ocean of Wang et al. (2016) for combined simulated MWI-ICI measurements. They used an approach similar to ours but with the difference of an additional frozen hydrometeor, different assumptions about the particle size distributions, and they used an additional neural network-

based classification before the retrieval. For snow they also assume slightly different scattering properties.

    Jiménez et al. (2007) conducted a simulated retrieval of IWP using channels similar to ISMAR and neural networks, but compared to our retrieval, they carried out the retrieval over land and for different meteorological situations. These authors defined the column integrated bulk mass of atmospheric ice as IWP, which, as written in the previous subsection, corresponds mostly with the SIWP of our retrieval. Comparing the MFE of SIWP with the retrieval error of IWP by Jiménez et al. (2007)

shows that their retrieval error is approximately half as large as the MFE of SIWP. One has to be cautious when comparing these errors, because the exact error definition in Jiménez et al. (2007) is not clear. Because the datasets and assumptions in Jiménez et al. (2007) differ from ours, compared to our retrieval, the errors cannot be expected to be the same, but they should be of the same order, which they are.

    A comparison with the error estimation of the SPARE-ICE product (Holl et al., 2014), which combines the results that were

obtained with the current operational microwave and infrared sensors, shows that the MFE of SIWP for $SIWP = 0.01\,\mathrm{kg\,m^{-2}}$ is of similar size as the MFE of IWP of the SPARE-ICE product and that with increasing SIWP the MFE of SIWP decreases to about half of the MFE of IWP of the SPARE-ICE product. The IWP of the SPARE-ICE product is defined as the column integrated bulk mass of atmospheric ice, but should be comparable to SIWP in our retrieval. For SIWP $< 0.01\,\mathrm{kg\,m^{-2}}$ the MFE of SIWP is larger than the MFE of IWP of the SPARE-ICE product. The SPARE-ICE product is a good measure to compare

with, because the SPARE-ICE product provides a good estimate of the performance of the latest operational passive sensors, but there are also two caveats in the comparison. Firstly, our MFE is based on model simulations under ideal conditions, whereas the MFE of SPARE-ICE is based on the 2C-ICE product (Deng et al., 2010), which is derived from lidar and radar measurements. Secondly, our error estimation is obtained from the perspective of the retrieval results, whereas that of Holl et al. (2014) is from the perspective of the reference data, but as long as the retrieval is offset-free this should not make a

significant difference. For SIWP $< 0.01\,\mathrm{kg\,m^{-2}}$ it is more effective to use a retrieval that includes thermal infrared channels as in SPARE-ICE, because the interaction between atmospheric ice and microwaves and submillimetre waves is too weak for such a low amount of SIWP, see Fig. 3. For now, we can keep in mind that our retrieval is capable of estimating SIWP with MFE lower than $100\%$ for SIWP$> 0.01\,\mathrm{kg\,m^{-2}}$, and that the MFE of SIWP is reduced to about $20\%$ for high SIWP.

    The MFE of LWP is of similar size as the MFE of RWP for LWP$< 0.1\,\mathrm{kg\,m^{-2}}$, but for LWP $> 0.1\,\mathrm{kg\,m^{-2}}$ the MFE of

LWP decreases to $30\%$. For LWP $> 0.05\,\mathrm{kg\,m^{-2}}$, the MFE is $< 100\%$. A MFE of $50\%$ for $LWP = 0.1\,\mathrm{kg\,m^{-2}}$ converted to

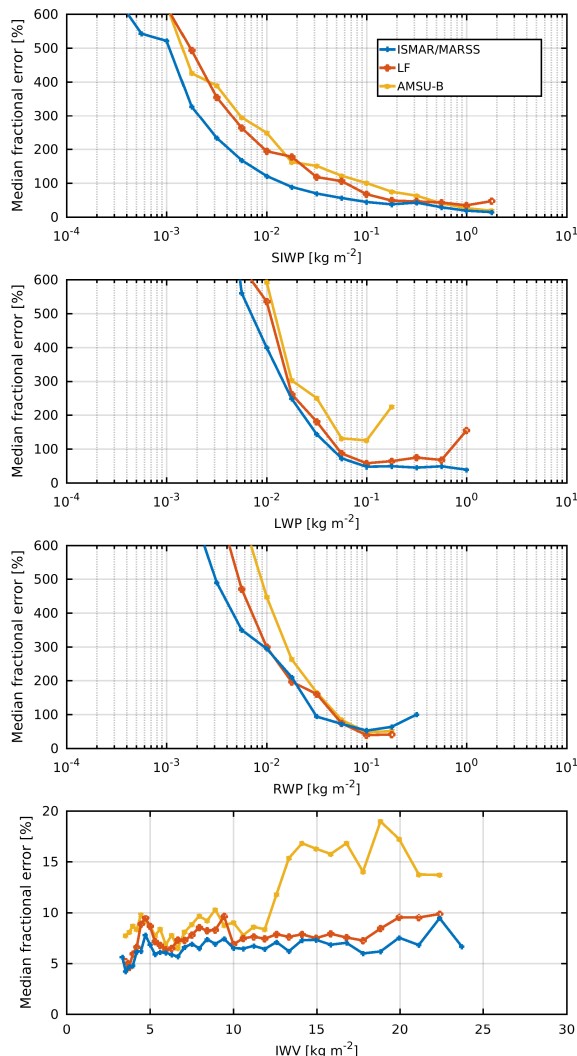

**Figure 5.** Median fractional error as function of the components of the retrieved state vectors. The blue lines indicate the MFE of the retrieval using all ISMAR and MARSS channels. The red lines indicate the MFE using all ISMAR and MARSS channels up to the 183 GHz channels for the retrieval and the orange lines indicate the MFE using a AMSU-B type sensor. See Sect. 4.3 for details.

absolute values approximately corresponds to an error of $0.05\,\mathrm{kg\,m^{-2}}$ and a MFE of $30\%$ for $LWP = 1\,\mathrm{kg\,m^{-2}}$ converted to absolute values approximately corresponds to an error of $0.3\,\mathrm{kg\,m^{-2}}$. English (1995) estimated an error of $0.03\,\mathrm{kg\,m^{-2}}$ to $0.05\,\mathrm{kg\,m^{-2}}$ for LWP $< 0.5\,\mathrm{kg\,m^{-2}}$ using a retrieval based on measurements of the $89\,\mathrm{GHz}$ channel and the $157\,\mathrm{GHz}$ channel of MARSS, which is of the same order as our retrieval. For LWP $> 0.5\,\mathrm{kg\,m^{-2}}$, English (1995) argued that the retrieval is

unreliable by estimating an error of $0.85\,\mathrm{kg\,m^{-2}}$ for a LWP of $1\,\mathrm{kg\,m^{-2}}$. However, these researchers performed their LWP retrieval on low liquid clouds, apparently without any ice. Compared to their results the error of our retrieval is almost one third of the error of their retrieval and the meteorological conditions in our retrieval are much more complicated. Horvath and Davies (2007) compared the retrieval of LWP of warm non-precipitating clouds from Tropical Rainfall Measurement Mission (TRMM) Microwave Imager (TMI) and from Moderate Resolution Imaging Spectroradiometer (MODIS). They found a RMS

difference $0.025\,\mathrm{kg\,m^{-2}}$ between the two LWP retrievals for a mean LWP of $0.1\,\mathrm{kg\,m^{-2}}$. Care needs to be exercised when comparing the errors of English (1995) and Horvath and Davies (2007) with our error estimate, because our error definition differs. Nonetheless, considering the fact that the meteorological conditions in our retrieval are more complex, because of coexisting frozen and liquid hydrometeors and because we do not focus on a specific cloud form, the estimated MFE of LWP is reasonable.

Our previous consideration of RWP indicated that the retrieval of RWP using MARSS and ISMAR is difficult. Except for the section around $RWP = 0.1\,\mathrm{kg\,m^{-2}}$, where the MFE of RWP is about $50\%$, the MFE of RWP is much larger than $100\%$. Interestingly, the MFE of RWP decreases for RWP $< 0.1\,\mathrm{kg\,m^{-2}}$ and afterwards the MFE increases with increasing RWP. If we compare the MFE of RWP with the offset of RWP, then we can identify the regions with the lowest MFE as the regions where the offset of RWP is zero. The MFE of RWP increases for RWP $> 0.1\,\mathrm{kg\,m^{-2}}$ with increasing RWP, because the offset

of RWP increases with increasing RWP even though the standard deviation of RWP changes little with increasing RWP. This is in contrast to the findings of Wang et al. (2016), who estimated a relative value of $< 40\%$ for RWP $> 0.1\,\mathrm{kg\,m^{-2}}$. The reason for this is that their training database includes more cases with higher RWP than our training database, so that their training database is more suitable for estimating RWP. If our database included more cases with higher RWP it is likely that our retrieval would provide a similar result to Wang et al. (2016).

Let us now consider the MFE of IWV, which is also shown in Fig. 5. As for the consideration of the offset of IWV, the MFE of IWV differs strongly from the results of the hydrometeor paths. The MFE of IWV is one order of magnitude smaller compared to the MFE of the hydrometeor paths and almost constant over the whole range of values changing little between $5\%$ to $8\%$. Converted to an absolute value, this corresponds to an error of $0.2\,\mathrm{kg\,m^{-2}}$ for low IWV and to an error of $2\,\mathrm{kg\,m^{-2}}$ for high IWV. This error range of IWV corresponds to the range of differences of several different IWV retrievals (microwaves,

infrared, radio sonde) and GPS retrieved IWV within the work of Buehler et al. (2012a). Note that, as we did not place any restriction on IWV, the retrieval for IWV is effective for cloudy conditions as well as for clear sky conditions.

### 4.3 Benefit of the high-frequency channels of ISMAR

It is interesting to explore the benefit of the new high-frequency channels of ISMAR. We answer this question by comparing the retrieval, which we name the "ISMAR-MARSS" retrieval hereafter, with two additional retrievals: one retrieval using all

channels up to $183\,\text{GHz}$ (Table 1, Ch. 1-10) and another retrieval using the $89\,\text{GHz}$, the $157\,\text{GHz}$, and the $183\,\text{GHz}$ channels (Table 1, Ch. 1, 7-10), which are the same 5 channels at which AMSU-B measures (Saunders et al., 1995), see also Sect. 2.2. We name the former and latter retrievals "LF" and "AMSU-B," respectively. Except for the number of channels used and the number of hidden layer neurons, the setup is exactly as for the ISMAR-MARSS retrieval. Compared to the ISMAR-MARSS retrieval the number of hidden layer neurons of the LF retrieval and the AMSU-B retrieval were reduced to reduce the chance of overfitting, but tests showed that this is still adequate. The LF and AMSU-B retrievals use 7 and 5 hidden layer neurons, respectively.

The MFE for each component of the state vector as a function of the corresponding component of the estimated state vector is shown in Fig. 5. The MFE for RWP of the LF retrieval and of the AMSU-B retrieval are shown only for the sake of completeness, because we already know from our above considerations that the retrieval is insufficient for RWP. Therefore, we concentrate on SIWP, LWP, and IWV. For SIWP, the MFE of the ISMAR-MARSS retrieval is reduced at $SIWP \approx 0.01\,\text{kg}\,\text{m}^{-2}$ below $100\%$, whereas the MFE of the LF retrieval and of the AMSU-B retrieval of SIWP decrease at $SIWP = 0.06\,\text{kg}\,\text{m}^{-2}$ and $SIWP = 0.1\,\text{kg}\,\text{m}^{-2}$ below $100\%$, respectively. At $SIWP = 0.06\,\text{kg}\,\text{m}^{-2}$ the MFE for the ISMAR-MARSS retrieval is already at $50\%$. For SIWP, the MFE of the LF retrieval and of the AMSU-B retrieval are consistently higher than the MFE of the ISMAR-MARSS retrieval, but with increasing SIWP the difference between the MFE decreases. Because of the higher frequencies of the ISMAR channels (Ch. 11-18) the MFE of SIWP can be reduced by a factor of as much as two with respect to the AMSU-B configuration. The $118\,\text{GHz}$ channels are less important for the retrieval of SIWP, because the difference between the LF retrieval and the AMSU-B retrieval is smaller.

For LWP, the MFE of the ISMAR-MARSS retrieval decreases monotonically, whereas the MFE of the LF retrieval and of the AMSU-B retrieval of LWP decreases only up to $LWP = 0.1\,\text{kg}\,\text{m}^{-2}$ and then increases with increasing LWP, whereas the MFE of the LF retrieval only increases slightly. The reason for the strong increase of the MFE of the AMSU-B retrieval is a strong underestimation of $LWP > 0.1\,\text{kg}\,\text{m}^{-2}$. The AMSU-B retrieval estimates almost no $LWP > 0.2\,\text{kg}\,\text{m}^{-2}$. The increase of the MFE of the LF retrieval is less strong than the MFE of the AMSU-B retrieval. The reason for the increase of the MFE of the LF retrieval is an increase of the offset with increasing LWP, which results in an overestimation of the LWP. Therefore, the higher frequency ISMAR channels (Ch. 11-18) deliver valuable information for the retrieval of LWP.

Thus, the lower frequency ISMAR channels (Ch. 2-6) and the higher frequency ISMAR channels (Ch. 11-18) are valuable for the retrieval of IWV. Whereas the MFE of IWV for the AMSU-B retrieval is on average about $10\%$ below an IWV of $12\,\text{kg}\,\text{m}^{-2}$ and higher than $10\%$ above an IWV of $12\,\text{kg}\,\text{m}^{-2}$, the MFE of IWV for the LF retrieval is on average approximately $8\%$ and the MFE of IWV for the ISMAR-MARSS retrieval is approximately $6\%$ on average.

Thus, we can say that compared to an AMSU-B type sensor, the ISMAR channels deliver very valuable information for the retrieval, especially for SIWP, but also for a more accurate IWV retrieval and for a LWP retrieval under complex meteorological conditions.

## 4.4 Basic Performance Summary

This section describes the retrieval tests under ideal conditions. This means, retrieval and test data are based on the same assumptions. By doing so, the error of the radiative transfer simulation and the error of the atmospheric model were excluded from this investigation. The investigated errors result from the artificial neural networks and from the physical limits of the retrieval, which are on the one hand the limited interaction between the electromagnetic radiation and the atmosphere and on the other hand the noise of the radiometers ISMAR and MARSS. Therefore, the investigated errors are an estimate of the limits of our retrieval approach. The retrieval error when applying the retrieval on measured brightness temperatures is likely to be larger, as the a priori assumptions will be never completely fulfilled.

One basic requirement of a retrieval is, in general, that the retrieval should be bias free or in our terms the retrieval should have no offset. Based on that, the retrieval fulfills this requirement for SIWP$> 0.01\,\mathrm{kg\,m^{-2}}$, LWP$> 0.1\,\mathrm{kg\,m^{-2}}$ and for IWV$>$ $3\,\mathrm{kg\,m^{-2}}$. If the retrieval has also an offset of zero for IWV$< 3\,\mathrm{kg\,m^{-2}}$, we cannot say, because there were almost no states with IWV$< 3\,\mathrm{kg\,m^{-2}}$. We can say that the requirement is not fulfilled for RWP.

In summary, a comparison with the simulated retrieval of Jiménez et al. (2007) showed that the performance of our SIWP is of the same order. The performance of our SIWP is also in good agreement with the performance of the SIWP retrieval of Wang et al. (2016). When the SIWP is not excessively small, above $10^{-2}\,\mathrm{kg\,m^{-2}}$ ISMAR has the potential to perform more effectively than the SPARE-ICE (Holl et al., 2014) product. For smaller SIWP, SPARE-ICE performs more effectively, because it uses infrared channels, which are more sensitive to very thin clouds than millimeter and sub-millimeter waves. A comparison with the retrieval of English (1995) and the study Horvath and Davies (2007) showed that the results of the LWP retrieval are reasonable. The LWP retrieval method is capable of retrieving LWP in situations with coexisting frozen and liquid hydrometeors. Furthermore, our retrieval is capable of retrieving IWV under cloudy and clear sky conditions with an error, which is comparable with existing clear-sky IWV retrievals.

A comparison of our retrieval with retrievals using only the channels up to $183\,\mathrm{GHz}$ enables us to conclude that the retrieval of SIWP strongly benefits from the higher frequency ISMAR channels (Ch. 11-18, see Table 1). The MFE of SIWP is reduced by a factor of two compared to retrievals using only channels up to $183\,\mathrm{GHz}$-channels. Both the IWV and LWP retrievals benefit from the higher frequency ISMAR channels.

## 5 Flight B897: Measurements on March 18, 2015

In this subsection, we describe the application of the retrieval to brightness temperatures measured during the FAAM flight B897 on March 18, 2015 as part of COSMICS. On that day, the FAAM BAe-146 aircraft measured a precipitating frontal system west of the coast of Iceland. The aircraft had several instruments on board to measure the size of ice particles, amongst which in-situ probes, ISMAR, and MARSS. We focus on the measurements of these two radiometers. Details about FAAM BAe-146 and the other instruments on board can be found on the website of FAAM (http://www.faam.ac.uk) .

Figure 6 shows the flight track, overlaid on MODIS images from March 18, 2015. The flight consisted of three north-south-transects across the frontal structure starting in the north. The transects were flown along a straight line starting at 66°N and

25°W and ending at 62°N and 25°W. The airplane required 2.5 h for the three transects. During these transects a total of 12 dropsondes (Vaisala Dropsonde RD94) were dropped. The altitude time series is also shown in Fig. 6. The airplane was above the clouds most of the time. During the flight the clouds varied from thin, broken clouds in the north to full-depth precipitating clouds in the south. The frontal structure moved slightly northwards during the flight.

Every time step at which the aircraft was not in stable straight and level flight was excluded from the brightness temperature time series to ensure that the retrieval is only applied to measurements recorded when the aircraft was at constant altitude with its wings level. In stable straight and level flight, the aircraft actually has a pitch of 5° resulting in slightly different incidence angle for ISMAR and MARSS instead of nadir, but this slight change in the incidence angle has no significant effect on the retrieval. The sampling period of the brightness temperature time series is 3.6 s. The time series is smoothed by a 3.5 min

running mean to improve the compatibility of the measurements with those of the ICON model and to reduce the amount of noise. A 3.5 min running mean corresponds to a path length of $\approx 23$ km. This is in the order of the smallest horizontal size of features that can be resolved within of the ICON model, which is twice the grid resolution of ICON. As stated in Sect. 3.4, different neural networks need to be trained for different flight altitudes. Thus, we divided the flight into nine discrete pressure levels, for each of which, neural networks, as described in Sect. 3.4, were trained using $6,000$ randomly selected

profiles from the database. These neural networks were applied to the measured brightness temperature time series, which is shown in Fig. 7. The flight consisted of three crossings of a frontal system. The brightness temperature time series starts at 12.3 h in the north, then flying southward till 13.4 h crossing the frontal system and flying back northward till 14.1 h and finally flying southward. The brightness temperature time series itself reflects the flight pattern, as it is symmetric around the turning points (13.4 h, 14.1 h). From the symmetry within the brightness temperature time series, it is to be expected that the

meteorological conditions are also symmetric with respect to the turning points. This symmetric pattern is a good test for the consistency of the retrieval procedure, because the retrieved hydrometeor path and IWV time series should reflect this pattern. In the 89 GHz channel, we can clearly see the crossing of the frontal system. At the beginning the brightness temperatures are about 190 K and this low brightness temperature indicates that the sensed radiation was emitted from the ocean surface. At 89 GHz the emissivity of the ocean surface is approximately 0.7, resulting in a brightness temperature of about 190 K for a

surface temperature of about 273 K. Over the ocean an increase in the amount of liquid water in the atmosphere leads to an increase in the brightness temperature at 89 GHz. When the aircraft moved towards the frontal system, the 89 GHz brightness temperature increased up to a maximum of 250 K around the turning point at 13.4 h. This increase in the brightness temperature enables us to conclude that there must be a strong increase in the amount of liquid water in the atmosphere, because the high brightness temperature indicates that the large amount of absorption suggests that the sensed radiation is not emitted by the

ocean surface but from somewhere in the lower troposphere.

## 5.1   Retrieval applied to Flight B897

Time series of the retrieved SIWP, LWP, and IWV are shown as blue lines in Fig. 8. In the absence of in-situ data except for the twelve water vapor profiles from dropsonde measurements, the retrieval is compared with the ICON model. The red lines indicate the value of the corresponding component of the ICON model state vectors interpolated to the time and location of the

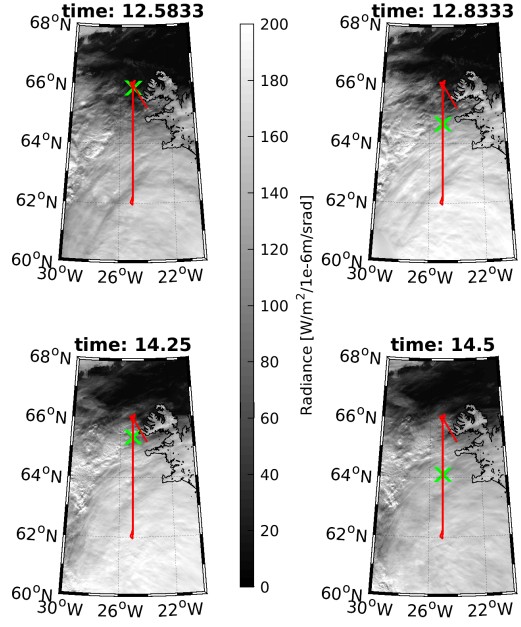

a)

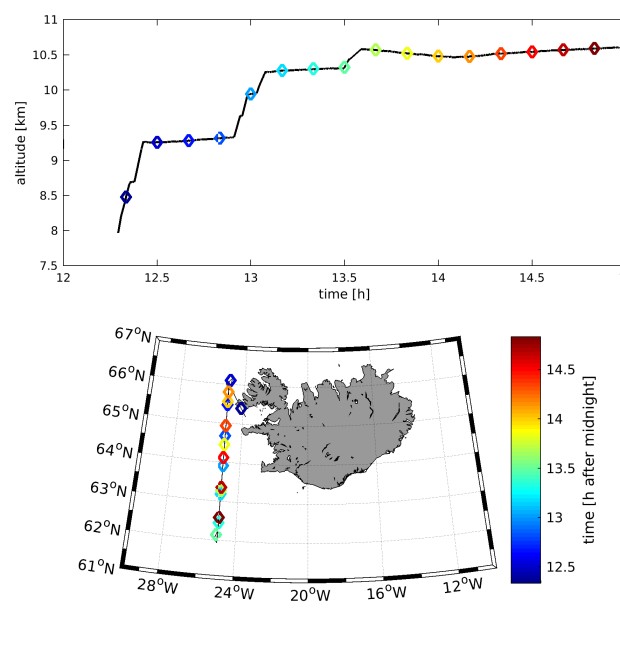

b)

**Figure 6.** (a) Images of MODIS (Terra and Aqua) band 1 (visible) during flight B897 on March 18, 2015 overlaid with the flight track (red lines) and the position of the aircraft (green crosses) at the MODIS measurement time. (b) Altitude time series of flight B897 and below map with the flight track (black) with a color marker indicating the time.

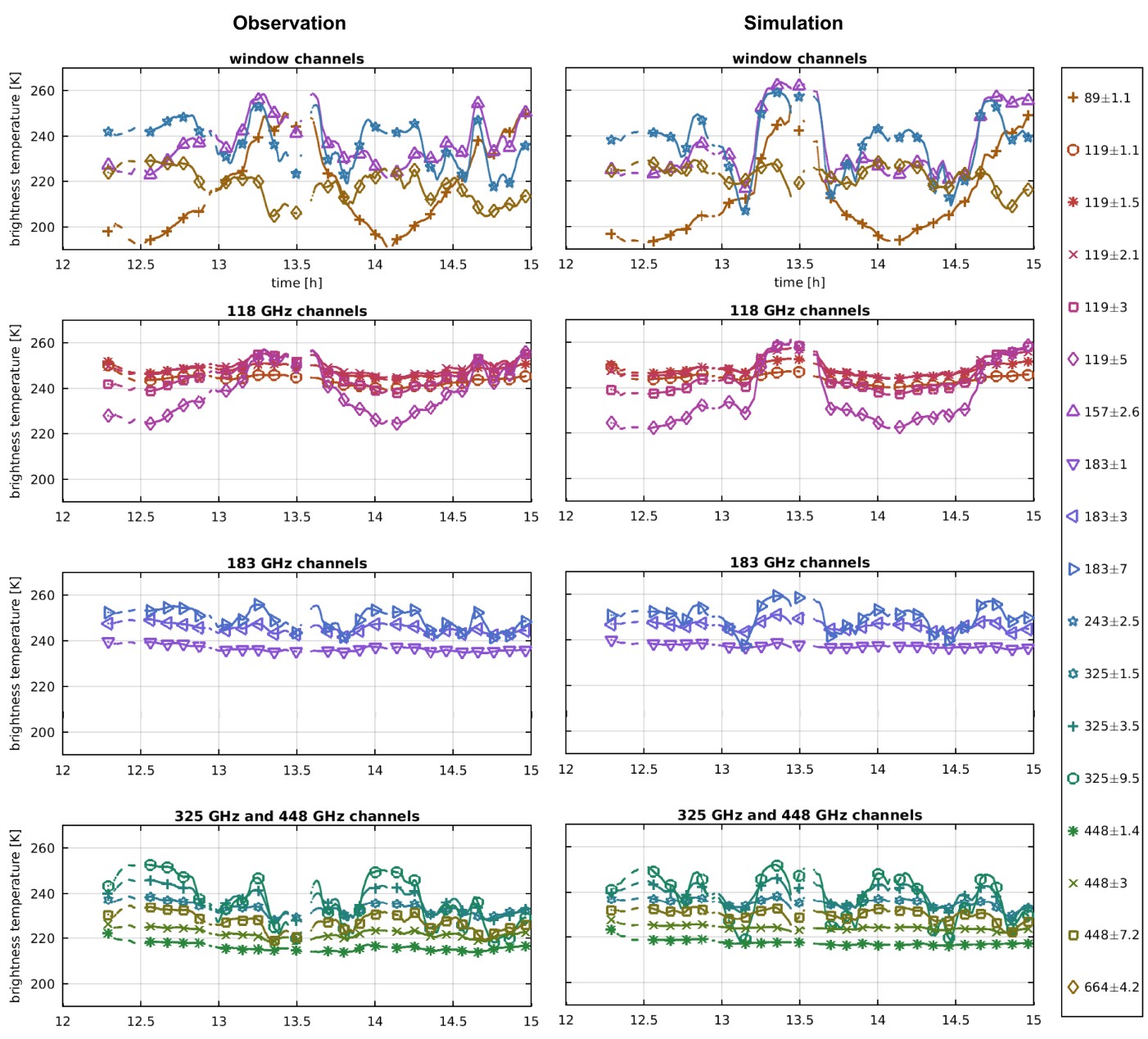

**Figure 7.** (left) observed brightness temperature time series of flight B897. (right) simulated brightness temperature time series of flight B897 The markers are shown only for every 100th value.

aircraft measurement. Of course, the ICON model itself is far from being perfect due to internal assumptions, limited temporal and spatial resolution. Therefore we cannot expect that the model is accurate in terms of retrieval quantities, time and location. To get an estimate for the uncertainty of the ICON model, we produced histograms of the corresponding components of the ICON state vectors within a $50\,\mathrm{km}$ radius and within $\pm 1\,\mathrm{h}$ of the time of measurement for every time step and location of the flight . The histograms are plotted as grey shades underneath the ICON time series. The model data itself is not considered as truth but it serves as a consistency check within this analysis.

Comparing the retrieval to the model state is not a true validation for several reasons, notably the dependence of the training data on the same model, and the fact that the model hydrometeors may be quite far from the true hydrometeors at the time and location of measurement. Nevertheless, testing whether the ICON simulations and ISMAR/MARSS measurements are comparable is important to ensure consistency, given our assumptions in representing the model hydrometeors in the radiative transfer simulations. Big errors in these assumptions would mean that the simulated and observed brightness temperature for a given profile would be very different. This implies that the result from the retrieval applied to the actual observation would be a very different to the model.

In general, the time series of the retrieved state vectors in Fig. 8 are within the given uncertainties and in reasonable agreement with the time series of the ICON model. The blue lines are mostly within the grayish area. The retrieved SIWP, LWP, and IWV time series are symmetric with respect to the turning points ($13.4\,\mathrm{h}$, $14.1\,\mathrm{h}$), which is consistent with the above-mentioned expectation. Although the agreement is good in general, there are substantial differences between the retrieval and the model, for example the time period between $13\,\mathrm{h}$ and $13.5\,\mathrm{h}$ of the SIWP time series. Possible sources for the difference between the retrieved time series and the modeled time series are:

1. The limit of the retrieval itself, namely the combined error from the neural network approach and the radiometer noise.

2. The assumptions for the radiative transfer simulations, namely the assumption about particle size distributions and hydrometeor types and their shape.

3. Misplacement. The ICON model can generally simulate the frontal structure during the flight; however, the ICON model cannot simulate the frontal structure exactly collocated in time and space of the measurements.

4. Unresolved features. the ICON model cannot simulate all the small details of the frontal structure that can be sensed by the aircraft measurements. Recall that, firstly, the airborne measurements have a much shorter sampling period ($T_{s,air} = 3.6\,\mathrm{s}$). Although the brightness temperature time series have been smoothed by a $3.5\,\mathrm{min}$ running mean, compared to the ICON model data ($T_{s,model} = 1800\,\mathrm{s} = 30\,\mathrm{min}$) it still captures more temporal variability. Secondly, the sampled space is much smaller than the grid size of the ICON model. The sampled space is of similar length in the along-track direction but in the across-track direction it is of the order of a hundred meters. In contrast, the ICON model has a grid resolution of about $10\,\mathrm{km}$ in the horizontal direction and $30\,\mathrm{min}$ in time, from which we interpolated the ICON model time series to the location and time of flight. Thus, we expect that the aircraft measurements sample more detailed features, which are not resolved in the ICON model.

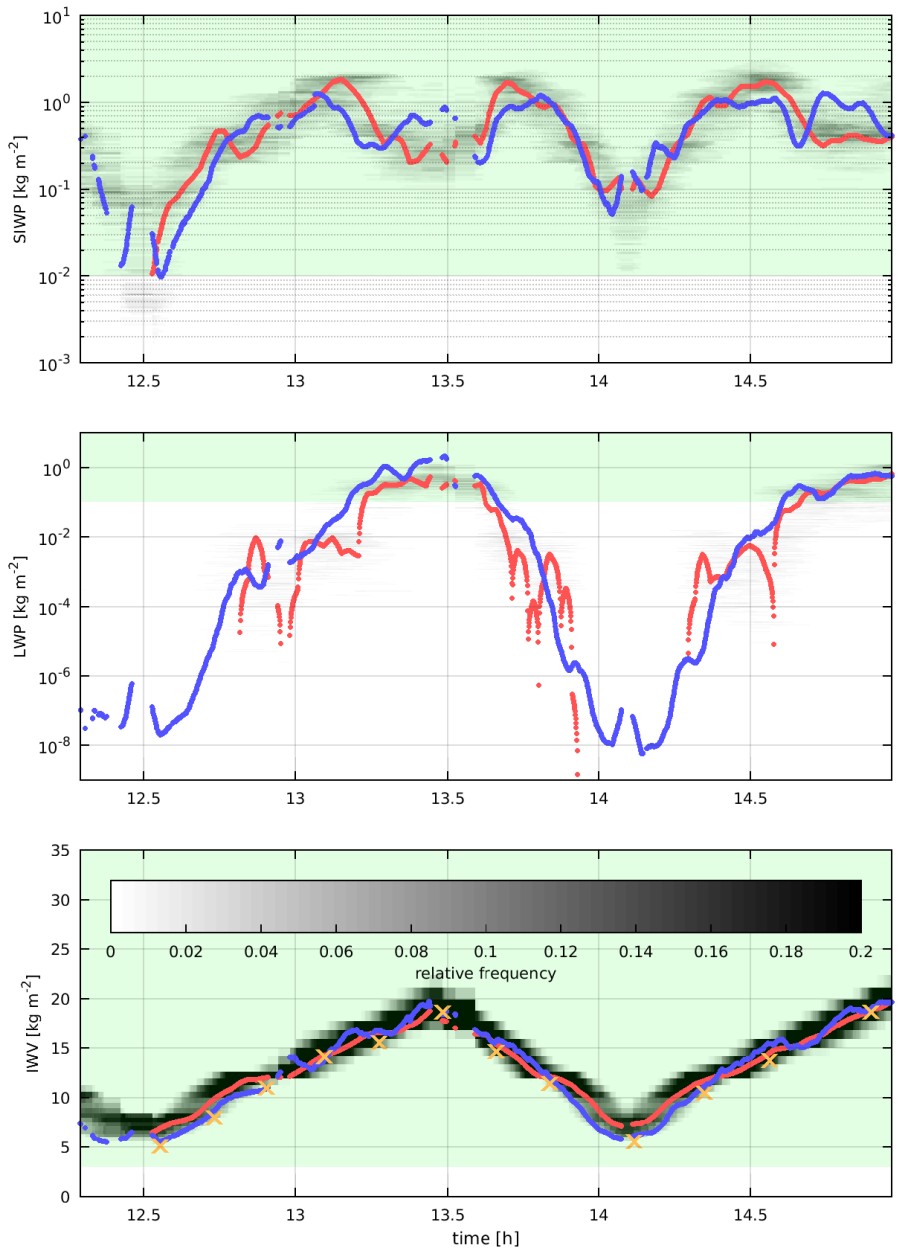

**Figure 8.** Time series of the retrieved SIWP, LWP, and IWV. The time series of the retrieved SIWP, LWP, and IWV are shown as blue lines. The time series of the ICON model SIWP, LWP, and IWV for the time and location of the FAAM during flight B897 are shown as red lines. Time series of histograms of the ICON model SIWP, LWP, and IWV are plotted as grey shades underneath, see text for details. The green shaded areas marks the range where the retrieval is offset free according to Sect. 4. The orange crosses indicate the IWV from the dropsonde measurements.

Time series of SIWP, LWP and IWV retrieved from simulated brightness temperatures of the flight are shown in Fig. 9 in a similar way as in Fig. 8 in order to illustrate the performance of the retrieval in idealized conditions. Under this ideal conditions, as simulation and retrieval are based on the same assumptions, the agreement between the retrieved time series and the model time series is very good, and differences are within the range expected from the analysis in Sect. 4. In Fig. 7 both the observed and simulated brightness temperature time series are shown. The observed brightness temperatures of the $89\,\mathrm{GHz}$ and the $118.75\,\mathrm{GHz} \pm 5\,\mathrm{GHz}$ channels show for example a steady increase between $12.5\,\mathrm{h}$ and $13.3\,\mathrm{h}$ flight time, whereas the increase of the simulated brightness is rather discontinuous with being flatter at the beginning and steeper after $13.2\,\mathrm{h}$. As described at the beginning of Sect. 5, an increase of the $89\,\mathrm{GHz}$ brightness temperature over ocean indicates an increase of liquid water within the atmosphere. The same holds for the $118.75\,\mathrm{GHz} \pm 5\,\mathrm{GHz}$ channel. The conclusion from this comparison of brightness temperatures is that in the model the increase of liquid water is delayed compared to reality. This implies that the model predicts the front further south, with a more rapid increase in liquid water. These behaviors are also reflected in the retrieved LWP time series from the observation (Fig. 8) and from the simulation (Fig. 9). LWP retrieved from the observation shows a more steady increase, whereas LWP retrieved from simulation shows a more discontinuous increase, with a strong increase at $13.2\,\mathrm{h}$. Therefore, it is unlikely that the differences arise from neural network and noise related uncertainties and that their effect is less important, because the retrieval shows for observation and simulation a coherent behavior in terms brightness temperature and LWP. Furthermore, the brightness temperature time series were smoothed to reduce the noise. However, it is likely that the differences mainly arise from the inaccuracies of the ICON model in the spatially, temporal, and structurally representation of the front, because the difference between LWP retrieved from the observation and LWP retrieved from simulation corresponds to the difference between observed and simulated brightness temperatures. Nonetheless, unresolved features in the ICON model cannot be excluded as possible source for the difference, too. The errors made by the radiative transfer simulations and the assumptions therein also influence the retrieval, but this reflects the general agreement between retrieval and model. A quantitative error estimate is difficult as there is no in-situ data to compare with and the model error of ICON and the radiative transfer simulations are unknown.

For IWV we can compare the retrieval with the in-situ data from the dropsondes. The dropsonde IWV is shown as orange crosses in Fig. 8. The retrieved IWV measurement captures the trend of the dropsonde IWV measurement, but compared to the dropsonde IWV the retrieved IWV is shifted to slightly higher values. The offset (mean difference) between the twelve dropsonde IWV values and the retrieved IWV value at the time of the start of dropsonde measurements is $0.5\,\mathrm{kg\,m^{-2}}$. This offset could be due to a dry bias of the radiosondes or due to a wet bias within the retrieval. Nonetheless, for an IWV value of $> 5\,\mathrm{kg\,m^{-2}}$ this offset results in an error of less than $10\%$. The rms difference between the twelve dropsonde IWV values and the corresponding retrieved IWV value is $0.8\,\mathrm{kg\,m^{-2}}$. This corresponds to an MFE of $16\%$ for an IWV value of $5\,\mathrm{kg\,m^{-2}}$ and to a MFE of $4\%$ for an IWV value of $18\,\mathrm{kg\,m^{-2}}$. When removing the offset, the rms difference is $0.6\,\mathrm{kg\,m^{-2}}$, which is similar to the random error $0.66\,\mathrm{kg\,m^{-2}}$ between the radiosonde measurements and the GPS measurement of the IWV values in Buehler et al. (2012a). The IWV error is in the expected range of Sect. 4.2. Despite the accuracy of the statistic being such that a detailed analysis is not possible, this comparison is encouraging, showing that the retrieval of IWV measurements, in general, is effective under both cloudy and clear sky conditions.

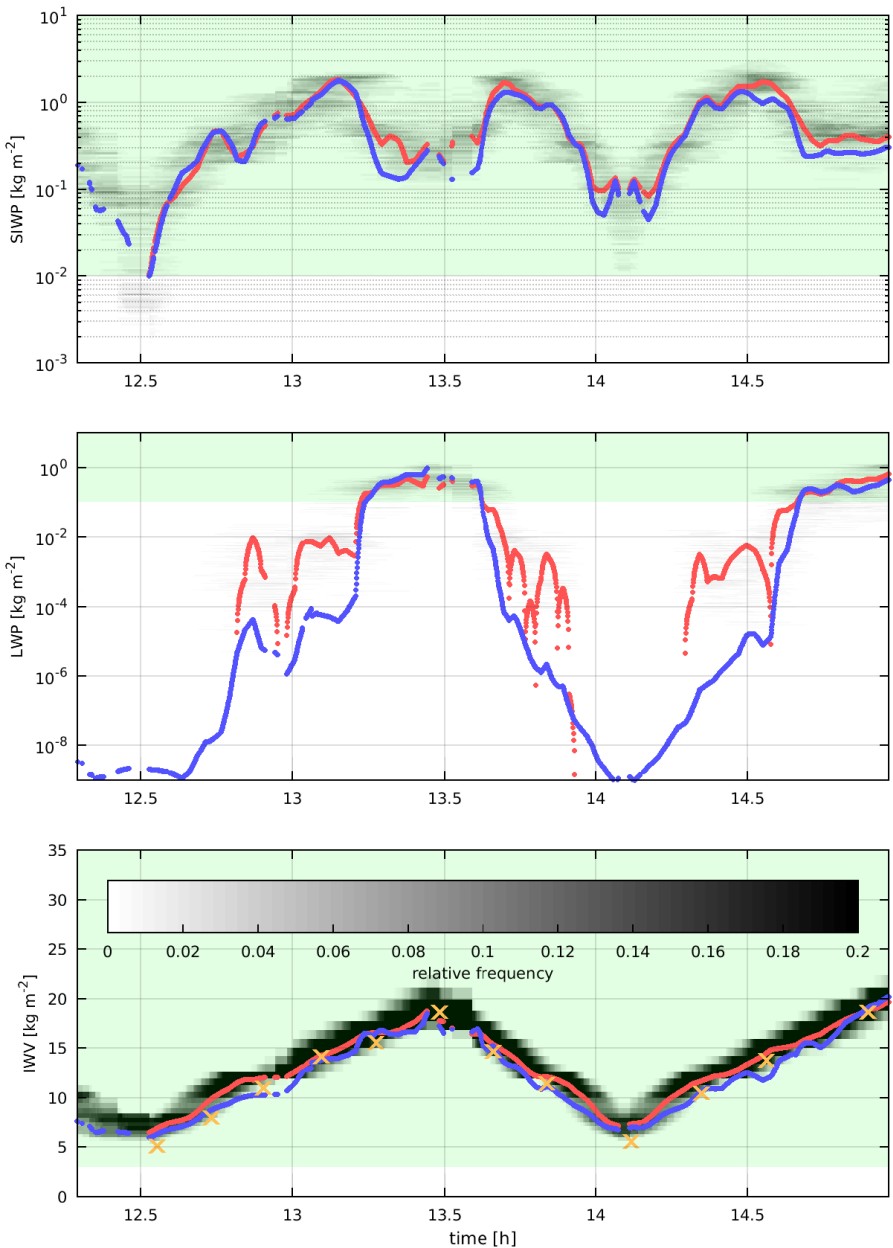

**Figure 9.** Time series of the retrieved SIWP, LWP, and IWV from simulated Flight. The time series of the retrieved SIWP, LWP, and IWV are shown as blue lines. The time series of the ICON model SIWP, LWP, and IWV for the time and location of the FAAM during flight B897 are shown as red lines. Time series of histograms of the ICON model SIWP, LWP, and IWV are plotted as grey shades underneath, see text for details. The green shaded areas marks the range where the retrieval is offset free according to Sect. 4. The orange crosses indicate the IWV from the dropsonde measurements.

We know from Sect. 4 that the retrieval is insufficient for RWP. Nonetheless, we apply the retrieval for RWP out of curiosity. Fig. 10 (top) shows the time series of the retrieved RWP, which seems to represent the general structure of the modeled time series. The retrieved RWP time series is symmetric with respect to the turning points (13.4 h, 14.1 h), which is consistent with the stated expectations. The retrieved RWP time series shows a strong increase within the time period between 12.5 h and 13.4 h with a maximum RWP at approximately 13.4 h, which is consistent with our conclusion from the brightness temperature time series. In Sect. 4, we concluded that the retrieval is insufficient for RWP, but at first glance the retrieval of RWP seems to be effective according to Fig. 10. We verified this by applying the retrieval to a simulated brightness temperature time series, because, if the retrieval of RWP was effective, then the retrieved RWP should be similar to the ICON RWP. The time series of the RWP retrieved from the simulated brightness temperature is shown in Fig. 10 (bottom). For RWP the blue and red lines are not in agreement. Therefore, our conclusion from Sect. 4 still holds. Even though the RWP retrieval is unreliable, it can still deliver some useful information, for example an approximate classification that indicates whether there is rain or not.

## 5.2 Summary of Flight Analysis

We applied the retrieval method to the brightness temperatures measured during Flight B897. As a consistency check we compared the retrieved state vectors with the ICON model state vectors, which we interpolated to the time and location of the aircraft measurements. Considering the given uncertainties, the agreement between the estimated SIWP, LWP, and IWV and the SIWP, LWP, and IWV from ICON is reasonable. There are strong local differences due to the misplacement of spatial features in the ICON model and small-scale variability. Compared to SIWP, LWP, and IWV, the RWP retrieval is less satisfactory, which is consistent with the results from Sect. 4. Furthermore, we compared the retrieved IWV with IWV from twelve dropsonde measurements. The mean difference between them is $0.5 \, \mathrm{kg \, m^{-2}}$ and the rms difference is $0.8 \, \mathrm{kg \, m^{-2}}$. We showed thereby, that we can estimate SIWP, LWP, and IWV with ISMAR in combination with MARSS.

## 6 Summary

This study involved an investigation of strategies for hydrometeor path retrieval from airborne radiometer measurements. We distinguish between cloud ice, which consists mainly of ice particles $< 100 \, \mu\mathrm{m}$, and snow, which consists mainly of ice particles $> 100 \, \mu\mathrm{m}$. This distinction between small and large ice particles is similar to the distinction in atmospheric models. We defined the CIWP as the column integrated bulk mass of cloud ice and we defined the SIWP as the column integrated bulk mass of snow. As the use of ISMAR and MARSS makes it possible to sense SIWP but not CIWP, we developed a retrieval method based on a neural network by using nadir viewing brightness temperature measurements with the main purpose of estimating SIWP. We also tried to estimate LWP, RWP, and IWV with the retrieval. The neural networks were trained by simulated brightness temperatures and atmospheric profiles from the ICON model. The brightness temperatures were simulated by ARTS with the atmospheric profiles from the ICON model as input. The scattering properties of the hydrometeors were assumed to behave as Mie spheres except for SIWP particles, which were assumed to behave like the aggregates from the Hong et al. (2009) database.

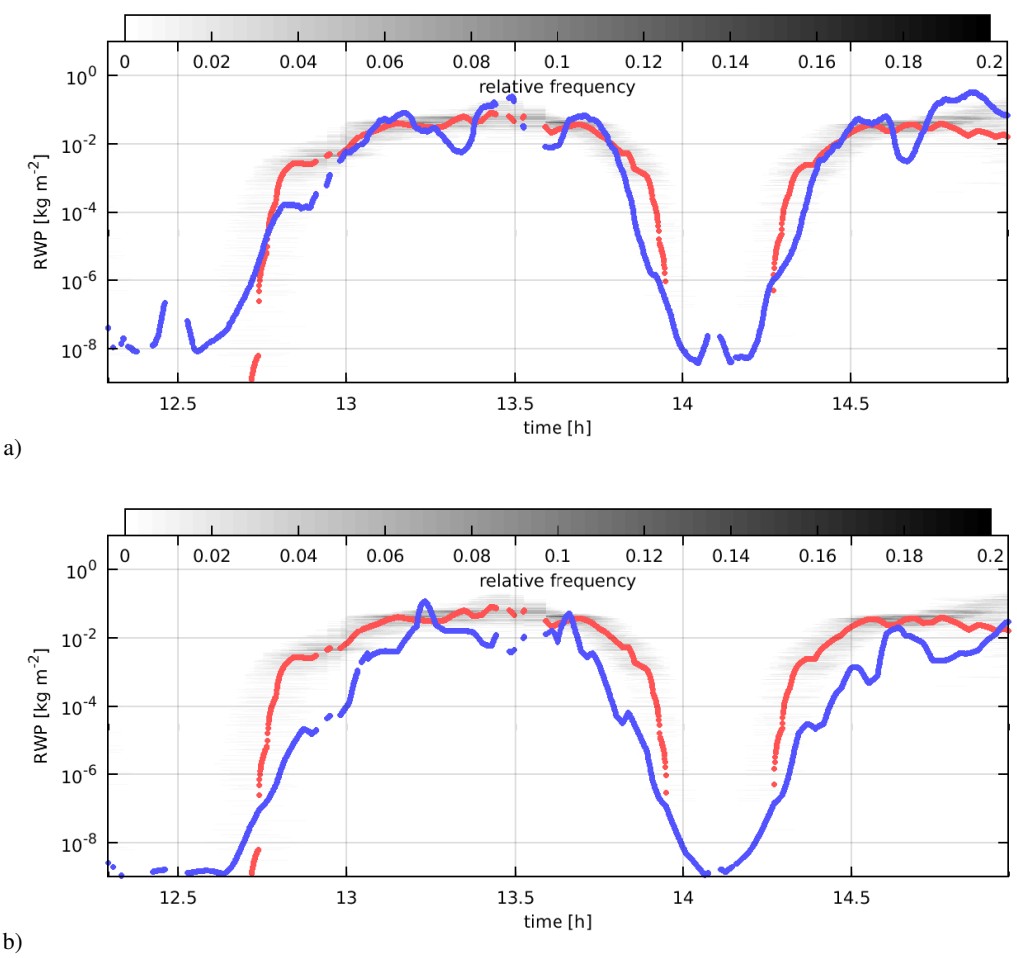

**Figure 10.** (a) Time series of the retrieved RWP. The time series of the retrieved RWP is shown as a blue line. The time series of the ICON model RWP for the time and location of the FAAM during flight B897 is shown as red line. Time series of histograms of the ICON model RWP are plotted as grey shades underneath, see text for details. (b) The same as in the top figure, but the RWP retrieved from the simulated flight.

We tested the retrieval with simulated measurements of which the true state is known. This test enabled us to estimate the physical limits of this retrieval process, which are:

- SIWP$> 0.01\,\mathrm{kg\,m^{-2}}$, then the MFE of our retrieval is lower than $100\%$, which decreases to about $20\%$ for high SIWP and the retrieval has an offset of zero.

- LWP$> 0.05\,\mathrm{kg\,m^{-2}}$, then the MFE of our retrieval is lower than $100\%$, which decreases to about $30\%$ for high LWP and the retrieval has an offset of zero.

- IWV$> 3\,\mathrm{kg\,m^{-2}}$, then the MFE is $5\%$ to $8\%$. Converted to an absolute value, this corresponds to an error of $0.2\,\mathrm{kg/m^2}$ for low IWV measurements and to an error of $2\,\mathrm{kg\,m^{-2}}$ for high IWV measurements.

The retrieval is insufficient for RWP determination, because it is not bias free and the MFE is mostly higher than $100\%$.

Furthermore, we showed that the magnitude of the error in the SIWP determination of the retrieval using ISMAR and MARSS measurements is only half of that of the retrieval using only AMSU-B channel combinations. This shows that estimating SIWP strongly benefits from sub-millimeter wave measurements, but also that estimating LWP and IWV benefits from the higher frequency ISMAR channels.

We applied the retrieval method to brightness temperature measurements recorded during Flight B897. As a consistency check we compared the estimated SIWP, LWP, and IWV values with the SIWP, LWP, and IWV values that were obtained by using the ICON model, which were interpolated to the time and location of flight B897. Considering the stated uncertainties, the agreement between the estimated SIWP, LWP, and IWV values and the SIWP, LWP, and IWV values obtained with ICON is reasonable. A comparison between the retrieved IWV values with those from the twelve dropsonde measurements shows that the mean difference between them is $0.5\,\mathrm{kg\,m^{-2}}$ and the rms difference is $0.8\,\mathrm{kg\,m^{-2}}$. We showed thereby, that we can use brightness temperature measurements obtained using ISMAR in combination with MARSS to estimate SIWP, LWP, and IWV. This is especially interesting in view of the upcoming METOP-SG mission, where ICI together with MWI will provide brightness temperature measurements with a similar combination of channels. Though, our retrieval is limited in season and latitude range, there is no fundamental limit in using neural network for global retrievals. The main requirement for global application is that the training database covers the wide range of global possible atmospheric conditions.

After establishing that the retrieval of SIWP, LWP, and IWV is effective, the next steps would be to firstly proceed beyond estimating integrated quantities and retrieve profiles, because of the considerable potential of the combination of the channels of ISMAR and MARSS, which we did not exploit in our actual retrieval. Secondly, the scattering properties of snow have to be investigated especially in the sub-millimeter range, because data for the scattering properties of this range of the electromagnetic spectrum are rare and partially inconsistent with measurements. The mass of the taken Hong aggregates is proportional to the third power of the maximum dimension of these aggregates (see also 3.2), whereas the measurements show that the mass is approximately proportional to the second power of the maximum dimension (Cotton et al., 2013). This is especially important in view of retrievals for the upcoming ICI sensor, because the retrieval results will strongly depend on the goodness of the scattering properties. Therefore, a more thorough validation is clearly needed, for example against in-situ measurements.

Setting up such validation experiments will be logistically challenging, ideally using at least two different aircraft, one with the radiometer and one with the in-situ probes. Co-located aircraft cloud radar would be also very helpful.

*Acknowledgements.* The authors would like to thank the crew and personnel involved in the COSMICS campaign, in particular Clare Lee and Stuart Rogers, who operated ISMAR and MARSS during flight B897. The BAe-146 research aircraft is operated by Airtask and Avalon and managed by the Facility for Airborne Atmospheric Measurements (FAAM), which is jointly funded by the Met Office and Natural Environment Research Council (NERC). The authors would like to thank Axel Seifert from the Deutscher Wetterdienst (DWD) for conducting and providing the ICON model simulations. Stefan A. Buehler was partially supported by DFG Research Group 1740 (contract BU2253/1-2), by BMBF project HD(CP)2 (contracts O1LK1502B and O1LK1505D), and by the DFG HALO research program (contract BU2253/3-1). Finally, our thanks to the ARTS radiative transfer community for their help with using ARTS.

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
