# Peer review of "Retrieval of an Ice Water Path over the Ocean from ISMAR and MARSS millimeter/submillimeter brightness temperatures"

_Atmospheric Measurement Techniques, 2017_

## Referee Comment (RC1) · Anonymous Referee #1 · 15 Aug 2017

Review comments for "retrieval of an ice water path over the ocean from ISMAR and MARSS millimeter/submillimeter brightness temperatures" by Brath et al.

This work details a neural-network retrieval algorithm that could retrieve some key hydrometer quantities at high accuracy from a combination of mm/sub-mm sensors. I especially very much like the sensitivity study using different combination of channels shown in Fig. 4, which quantitatively showcases us the advantage of using combined mm/sub-mm channels. With the nice spread of water vapor channels at 183, 325 and 448 GHz during the flight, I believe a further realistic (and not too ambitious) goal is to actually retrieve the entire water vapor vertical profile, but this is beyond the scope of

this study (but giving people very much hope).

The writing is not concise, but very clear, detailed and easy to follow. The comparisons against previous works and discussions are comprehensive. The scientific value of this work is high. I fully support the final publication of this work on AMT.

There are five major comments that I'd recommend the authors to address/discuss in the revision: 1) The training database and the neural network was generated from the simulated TB and ICON model simulated atmospheric profiles for the same frontal case for Flight B897. Later on, the retrieved physical quantities are then evaluated against ICON model simulated ones again, the latter of which is treated as the "truth". Under this logic flow, the discrepancies between them are hard to justify whether they can be attributed to (1) the imperfection of ICON model simulation; (2) the imperfection of ARTS model; or (3) the imperfection of the neural network. I understand that SWP/LWP/RWP are very difficult to measure during field campaign, and the only measurable "truth" IWV compares well, which increases the credibility of the retrieval. But authors need to be explicit of the caveat of the whole underlying logic of building-up your retrieval system.

2) There is no error analysis in this paper at all. No discussion about the error sources of the retrieved results either. The errors need to be addressed in the revision.

3) Sort of following my major comment#1, one way to show the credibility of your ICON model simulation and the accuracy of ARTS calculation is to show the simulated TB for each channel on top of Fig. 6 (maybe increase the number of panels but show simulated and observed TB lines one-to-one). In that way, at least we can exclude or conclude the imperfection of neural network model as the major source for some discrepancies.

4) Are the definitions of ice, snow, liquid and rain, including the size, shape and density are likely to be inconsistent between ICON model and ARTS, correct? What are the drawbacks of these inconsistencies, should they exist?

5) The "no-offset" conclusion draw from Fig. 3 is more or less biases when SIWP/LWP/RWP is very small, as there's a tight linear line across the upper-domain that "balances" the broad "under-estimation" domain. That means for small values that ISMAR+MARSS are not sensitive to, this retrieval approach tends to generate "bi-model" solutions that not behave symmetrically. Rather than arguing that the "offset is zero", I'd suggest you consider drop trusting the small SWP/LWP/RWP values, and mark your thresholds explicitly on Fig. 7. In this way, readers won't bother thinking why ICON simulated LWP "oscillates" against retrieved ones when the flight enters and leaves the fronts, as the retrieved LWP are so low that the retrieved value itself is not very meaningful.

In addition, some minor comments need to be clarified: 1) In the title, would "snow water path" be more accurate than "ice water path"?

2) For the viewing geometry of ISMAR and MARSS, what does "nadir angle" mean? Do they share the same viewing geometry with GMI/CoSSIR? How do you deal with different foot-print size and beam filling effect of different channels in ARTS?

3) 100um threshold used to separate snow and ice particles are more or less too arbitrary. I'm not familiar with the set-up with ICON model. As the microphysics schemes is not explicitly presented in the current manuscript (e.g., two-moment or one moment? How many hydrometer species? Do you allow super-cooled water to present? What's the ice/snow density? etc.), I'm not sure if the definition of SWP is consistent of what's been defined in the ICON model simulations.

---

## Referee Comment (RC2) · Anonymous Referee #2 · 24 Sep 2017

This study investigates high frequency microwave and sub-mm based retrievals of a various quantities like snow water path, rain water path, and integrated water vapor using a neural network (NN) retrieval methodology. My overall impression of this manuscript is that is extremely well written, thorough, and advances the state of retrieval science in the sub-mm portion of the electromagnetic spectrum. The authors included relevant and necessary discussion sections that elucidate major sources of uncertainty. I was admittedly thinking of possible major concerns, but authors inevitably preemptively addressed these concerns with thorough discussion sections within the manuscript. I truly appreciate these efforts by the authors. I suggest minor revisions based on the comments below, mostly related to NN methodology and applicability to

[Figure]

eventual global retrievals.

Page 6, Lines 4-5: "Because the atmospheric profiles were from the same season and the same region as the measurements, these profiles are expected to sufficiently cover the situations encountered during the measurement flight." The NN training dataset serves as arguably the most important component of this retrieval methodology. Populating the training dataset with a sufficient number of representative profiles is absolutely crucial for success. My question: this study analyzes a case study, comparing it to airborne observations, so the NN training dataset is probably sufficient and representative. But this particular region is susceptible to cold air outbreaks and synoptic scale weather systems that ultimately drive the weather and associated cloud formations. I assume the three simulated days chosen have representative synoptic conditions of this region, including a representative cold air outbreak scenario? This seems especially important for sample field campaign airborne retrievals presented later in the manuscript. Is a NN feasible for global retrievals? How do you sufficiently populate a database for global retrieval applications? These final two questions are relevant if the authors want to utilize this methodology for eventual ICI retrievals. Perhaps this question might be best addressed in the Summary section.

Page 6 Lines 14-15: "No explicit spectral response function was used to simulate the the ISMAR and MARSS channels; instead, we conducted monochromatic radiative transfer simulations for the center frequencies of the two side bands of each channel and obtained their average". This seems like a reasonable approach. Can the authors supply any sample uncertainties for this methodological approach? I assume uncertainties may increase under highly scattering conditions for sub-mm frequencies with weighting functions low enough in the atmosphere to be prone to ice scattering.

Page 6, Near line 25: FASTEM discussion regarding surface emissivity – can simple examples be provided that illustrate the lack of surface sensitivity for a few representative sub-mm channels? This type of analysis could be appropriate as a supplement, or at least provide references if this type of work has been published beforehand.

Page 6, last paragraph: Suggest adding unit information parenthetically. Current sentence seems awkward with comma devoted to unit information. For example, "cloud ice water (converted to kg m-3)". Or just include the units parenthetically without "converted to" wording. Also verify AMT publication standards regarding unit display. For example, kg/m3 versus kg m-3.

Page 7, Cloud ice: I am curious why a different Hong et al (2009) particle rendition is not chosen for cloud ice scattering simulations? I understand soft spheres may be fine for very small ice particles when the size ratio is sufficiently small, but do some sub-mm channels violate this small size ratio restriction and necessitate using DDA databases instead of soft spheres?

Page 12, Lines 19-25: I appreciate the authors being frank with possible downsides of using neural networks. Not having used NN before, is computational burden also excessive? It seems like the combined computational burden of (a) needing a large sample of numerical model results to populate the training dataset and (b) adopting an ensemble NN approach make this exercise fairly computationally intensive. This approach seems defensible and justifiable for the current application of illustrating retrieval efficacy for various parameters using combined microwave and sub-mm channels. But will a NN approach be untenable for real-time retrievals from space borne sensors (for instance, when ICI and MWI are eventually launched)?

Page 22, lines 5-6: Any specific reason why a 3.5 minute running average was chosen versus a different time averaging duration?

Section 5.1: So the NN training datasets from all 3 numerical simulations were used for the retrievals shown in this section? Or was the training dataset from the 18 March simulation results applied exclusively to this case? This is not a major issue for the results presented in this particular study, but my main question is how a similar NN retrieval methodology can be applied to a global dataset. Would the training dataset require daily simulations to provide a robust training dataset for global retrievals? Or

would a handful of simulations that temporally and strategically sampled various seasons suffice?

Summary section: As mentioned, ice scattering simulations are rare for sub-mm frequencies. I would add more emphasis in this section (another sentence or two) to encourage the community to produce more ice model scattering datasets at frequencies exceeding 200 GHz. This seems like a necessary research step to improve ice and snow column retrievals at sub-mm frequencies.

---

## Author Comment (AC1) · 21 Oct 2017

**Answers to Anonymous Referee #1**

October 21, 2017

Reviewer:

Review comments for "retrieval of an ice water path over the ocean from ISMAR and MARSS millimeter/submillimeter brightness temperatures" by Brath et al. This work details a neural-network retrieval algorithm that could retrieve some key hy- drometer quantities at high accuracy from a combination of mm/sub-mm sensors. I especially very much like the sensitivity study using different combination of channels shown in Fig. 4, which quantitatively showcases us the advantage of using combined mm/sub-mm channels. With the nice spread of water vapor channels at 183, 325 and 448 GHz during the flight, I believe a further realistic (and not too ambitious) goal is to actually retrieve the entire water vapor vertical profile, but this is beyond the scope of this study (but giving people very much hope). The writing is not concise, but very clear, detailed and easy to follow. The comparisons against previous works and discussions are comprehensive. The scientific value of this work is high. I fully support the final publication of this work on AMT.

There are five major comments that I'd recommend the authors to address/discuss in the revision:

**1) The training database and the neural network was generated from the simulated TB and ICON model simulated atmospheric profiles for the same frontal case for Flight B897. Later on, the retrieved physical quantities are then evaluated against ICON model simulated ones again, the latter of which is treated as the "truth". Under this logic flow, the discrepancies between them are hard to justify whether they can be attributed to (1) the imperfection of ICON model simulation; (2) the imperfection of ARTS model; or (3) the imperfection of the neural network. I understand that SWP/LWP/RWP are very difficult to measure during field campaign, and the only measurable "truth" IWV compares well, which increases the credibility of the retrieval. But authors need to be explicit of the caveat of the whole underlying logic of building-up your retrieval system.**

Answer:

The training database and the neural networks were not specifically generated from simulated TB and ICON model simulated atmospheric profiles for the same frontal case for Flight B897. We agree that the text was maybe misleading. We added an additional figure (Fig. 1) in Sect 3.1, where the position and time of each randomly selected profile is shown. We added additional sentences (p. 6 line 10 to p. 7 line 4) , which explain that the database covers a much wider range of atmospheric conditions and that it is not optimized for that specific flight.

We do not consider the ICON as "truth", but this was maybe not emphasized enough. We revised the introduction (p. 3 lines 8-25), which now includes the basic idea of the paper to emphasize the logic flow of the paper. We additionally revised Sect. 5.1 (p. 23 line 32 to p. 26 line 13) now explicitly saying that the ICON model data is not considered as truth and that the comparison is considered as a consistency check. We also address now explicitly the imperfection of the ICON model. Furthermore, we included in Sect. 5.1 (p. 26 line 18 to p. 28 line 23) a more detailed discussion

and analysis of possible reasons for the differences between the retrieval and model including one modified figure (Fig. 7) and an additional figure (Fig. 9).

Reviewer:

**2) There is no error analysis in this paper at all. No discussion about the error sources of the retrieved results either. The errors need to be addressed in the revision.**

Answer:

We do not agree, that there is no error analysis at all. In Sect. 4 we comprehensively address the combined error due to the neural networks and the noise of the radiometer. But we agree, that it was not stated explicit enough and that the discussion in Sect 5.1 was not detailed enough. We revised the beginning of Sect. 4 (p. 14 line 24 to p. 15 line 2) now explicitly saying, that the addressed error is due to the neural networks and the noise of the radiometer, that modeling errors are excluded in Sect. 4 and that the retrieval errors are likely to be larger when applied to observations. According to that, we adapted the summary of Sect. 4 (p. 22 lines 13-21). Furthermore, see also previous answer, we included in Sect. 5.1 (p. 26 line 18 to p. 28 line 23) a more detailed discussion and analysis of possible reasons for the differences between the retrieval and model arguing why the differences result mainly from model imperfections and not from the neural networks.

Reviewer:

**3) Sort of following my major comment#1, one way to show the credibility of your ICON model simulation and the accuracy of ARTS calculation is to show**

**the simulated TB for each channel on top of Fig. 6 (maybe increase the number of panels but show simulated and observed TB lines one-to-one). In that way, at least we can exclude or conclude the imperfection of neural network model as the major source for some discrepancies.**

Answer:

We followed the suggestion and included the simulated brightness temperatures in Fig. 7. We further used the simulated brightness temperatures in the revised analysis and discussion of in Sect. Sect. 5.1 (p. 26 line 18 to p. 28 line 23) .

Reviewer:

**4) Are the definitions of ice, snow, liquid and rain, including the size, shape and density are likely to be inconsistent between ICON model and ARTS, correct? What are the drawbacks of these inconsistencies, should they exist?**

Answer:

Yes, the definitions of ice, snow, liquid and rain, in terms of size, shape and density are inconsistent between ICON model and ARTS. But this is not problematic, because the function of the ICON model for the database is simply to deliver physically realistically profiles, which span the range of conditions that may be encountered. For this case, it is not needed to be consistent with the ICON model. If we would be interested in the microphysics of the ICON model than consistency would be needed. We added similar sentences to Sect. 3.2 (p.8 lines 27-31).

Reviewer:

**5) The "no-offset" conclusion draw from Fig. 3 is more or less biases when SIWP/LWP/RWP is very small, as there's a tight linear line across the upper-domain that "balances" the broad "under-estimation" domain. That means for small values that ISMAR+MARSS are not sensitive to, this retrieval approach tends to generate "bi-model" solutions that not behave symmetrically. Rather than arguing that the "offset is zero", I'd suggest you consider drop trusting the small SWP/LWP/RWP values, and mark your thresholds explicitly on Fig. 7. In this way, readers won't bother thinking why ICON simulated LWP "oscillates" against retrieved ones when the flight enters and leaves the fronts, as the re-trieved LWP are so low that the retrieved value itself is not very meaningful.**

Answer:

We agree. We added in Fig. 8 and in the newly added Fig. 9 marks, which indicate the range, where the retrieval is considered as offset free according to Sect. 4. We added some words and a sentence calling the specific behavior bimodal (p. 15 lines 23-24).

Reviewer:

In addition, some minor comments need to be clarified:

**1) In the title, would "snow water path" be more accurate than "ice water path"?**

Answer:

We agree, it would be more accurate to use the term *"snow water path"* instead of *"ice water path"*, but within the literature the term "ice water path" is more familiar. Furthermore, as mentions in Sect. 4, within the literature the term *"ice water path"* is frequently used for total amount of frozen hydrometeors, which mainly consist in our study of snow. Therefore, we think, it is better to use the more familiar term in the title and distinguish in the text.

Reviewer:

**2) For the viewing geometry of ISMAR and MARSS, what does "nadir angle" mean? Do they share the same viewing geometry with GMI/CoSSIR? How do you deal with different foot-print size and beam filling effect of different channels in ARTS?**

Answer:

We revised Sect. 2 explaining the viewing geometry of ISMAR (p. 4 lines 10-17) and MARSS (p. 4 line 21) thoroughly. The nadir +50 degree view of ISMAR is designed to give a close match in incidence angle to conically-scanning imagers such as ICI, GMI and CoSSIR as used during the Tropical Composition, Cloud and Climate Coupling (TC4) experiment in 2007 (Evans et al., 2012).

We neglect possible effects due to different footprint sizes and beam-filling as the foot-print of MARSS and ISMAR are much smaller than a grid cell from the ICON model. The footprint size at ground level is pretty much the same for all the ISMAR channels and are in the order of $700 \, \text{m}$ for a flight altitude of $10 \, \text{km}$. The footprint sizes of MARSS are in the order of $1,500 \, \text{m}$ for the $89.0 \, \text{GHz}$ and the $157.05 \, \text{GHz}$ channels and in the

order of $1,000 \, \text{m}$ for the $183.31 \, \text{GHz}$ channels. The footprints of ISMAR and MARSS are much smaller than footprints from similar satellite instruments. We added a similar statement to Sect. 3.2 (p.7 lines 17-20).

Reviewer:

**3) 100um threshold used to separate snow and ice particles are more or less too arbi- trary. I'm not familiar with the set-up with ICON model. As the microphysics schemes is not explicitly presented in the current manuscript (e.g., two-moment or one moment? How many hydrometer species? Do you allow super-cooled water to present? What's the ice/snow density? etc.), I'm not sure if the definition of SWP is consistent of what's been defined in the ICON model simulations.**

Answer:

The $100 \, \mu\text{m}$ threshold is not set arbitrary but emerges from the used size distributions. We agree that this point was not stated clearly. We added a few sentences in the introduction (p. 2 line 4) and in Sect. 3.2 (p. 9 lines 20-21).

The ICON runs used a 1-moment microphysics scheme with four distinct hydrometeor types namely liquid cloud water, cloud ice, rain and snow, which is also mentioned in the revised text (p. 8 lines 13-14) , now. Super cooled water exist in the model runs. The microphysics used in the paper differs in terms of size, shape, and density from the ICON internal but in the basic definition they are the same.

**References**

KF Evans, JR Wang, O'C Starr, G Heymsfield, L Li, L Tian, RP Lawson, AJ Heymsfield, A Bansemer, et al. Ice hydrometeor profile retrieval algorithm for high-frequency microwave radiometers: application to the cossir instrument during tc4. *Atmospheric Measurement Techniques*, 5(9):2277–2306, 2012.

---

## Author Comment (AC2) · 21 Oct 2017

**Answers to Anonymous Referee #2**

October 21, 2017

Reviewer:

This study investigates high frequency microwave and sub-mm based retrievals of a various quantities like snow water path, rain water path, and integrated water vapor using a neural network (NN) retrieval methodology. My overall impression of this manuscript is that is extremely well written, thorough, and advances the state of retrieval science in the sub-mm portion of the electromagnetic spectrum. The authors included relevant and necessary discussion sections that elucidate major sources of uncertainty. I was admittedly thinking of possible major concerns, but authors inevitably preemptively addressed these concerns with thorough discussion sections within the manuscript. I truly appreciate these efforts by the authors. I suggest minor revisions based on the comments below, mostly related to NN methodology and applicability to eventual global retrievals.

**Page 6, Lines 4-5: "Because the atmospheric profiles were from the same season and the same region as the measurements, these profiles are expected to sufficiently cover the situations encountered during the measurement flight." The NN training dataset serves as arguably the most important component of this**

**retrieval methodology. Populating the training dataset with a sufficient number
of representative profiles is abso- lutely crucial for success. My question: this
study analyzes a case study, comparing it to airborne observations, so the NN
training dataset is probably sufficient and rep- resentative. But this particular
region is susceptible to cold air outbreaks and synoptic scale weather systems
that ultimately drive the weather and associated cloud formations. I assume the
three simulated days chosen have representative synoptic condi- tions of this
region, including a representative cold air outbreak scenario? This seems espe-
cially important for sample field campaign airborne retrievals presented later in
the manuscript. Is a NN feasible for global retrievals? How do you sufficiently
populate a database for global retrieval applications? These final two questions
are relevant if the authors want to utilize this methodology for eventual ICI re-
trievals. Perhaps this question might be best addressed in the Summary section.**

Answer:

The three simulation days were originally chosen to cover three different FAAM flights.
We did not check if there was a cold air outbreak scenario in the model runs. We know
that in reality above 70°N overnight between the 18th and 19th March cold-air outbreak
conditions existed. Therefore, we assume, that some profiles are included. We added
an additional figure (Fig. 1) in Sect 3.1, where the position and time of each randomly
selected profile is shown.

In general neural network retrievals are feasible for global retrievals. For example,
Holl et al. (2014) used neural networks to retrieve ice water path. The crucial point for
neural network retrievals is that the database covers the wide range of globally possible
atmospheric conditions. By using for example, additional ICON model runs for several
globally distributed regions and different seasons, our retrieval can be expanded to
global applications. We added a similar statement in Sect. 3.4 (p. 14 lines 16-22).

Reviewer:

**Page 6 Lines 14-15: "No explicit spectral response function was used to simulate the the ISMAR and MARSS channels; instead, we conducted monochromatic radiative transfer simulations for the center frequencies of the two side bands of each channel and obtained their average". This seems like a reasonable approach. Can the authors supply any sample uncertainties for this methodological approach? I assume uncer- tainties may increase under highly scattering conditions for sub-mm frequencies with weighting functions low enough in the atmosphere to be prone to ice scattering.**

Answer:

We now provide some sample uncertainties in the text (p. 7 lines 15-17). It is unlikely, that these uncertainies under highly scattering conditions increase, because the change of the scattering properties over the range of half bandwith is small. Furthermore, as the number of scatterer sizes are limited, the interpolation uncertainty is likely to be bigger than the uncertainties by using only one frequency per pass band.

Reviewer:

**Page 6, Near line 25: FASTEM discussion regarding surface emissivity – can simple examples be provided that illustrate the lack of surface sensitivity for a few representa- tive sub-mm channels? This type of analysis could be appropriate as a supplement, or at least provide references if this type of work has been published beforehand.**

Answer:

We now provide some simple examples for the $243.3$ GHz and the $325.15$ GHz $\pm 9.5$ GHz channel in Sect. 3.2 (p. 7 line 29 to p. 8 line 8).

Reviewer:

**Page 6, last paragraph: Suggest adding unit information parenthetically. Current sen- tence seems awkward with comma devoted to unit information. For example, "cloud ice water (converted to kg m-3)". Or just include the units parenthetically without "converted to" wording. Also verify AMT publication standards regarding unit display. For example, kg/m3 versus kg m-3.**

Answer:

We changed it to "cloud ice water in $\mathrm{kg\,m^{-3}}$" and we adapted the unit display in the text.

Reviewer:

**Page 7, Cloud ice: I am curious why a different Hong et al (2009) particle rendition is not chosen for cloud ice scattering simulations? I understand soft spheres may be fine for very small ice particles when the size ratio is sufficiently small, but do some sub-mm channels violate this small size ratio restriction and necessitate using DDA databases instead of soft spheres?**

Answer:

You are right in that for that case it would be better to use DDA based scattering properties and it can be that some sub-mm channels violate for some particle sizes this small size ratio restriction, but as the overall scattering of cloud ice is very small even for high frequencies (see Fig. 3a) and as the usage of DDA based scattering properties will not substantially change this, it is not important. Therefore, soft spheres are fine in our case. When using a different size distribution, which provides larger particle sizes, your point comes into play.

Reviewer:

**Page 12, Lines 19-25: I appreciate the authors being frank with possible downsides of using neural networks. Not having used NN before, is computational burden also excessive? It seems like the combined computational burden of (a) needing a large sample of numerical model results to populate the training dataset and (b) adopting an ensemble NN approach make this exercise fairly computationally intensive. This approach seems defensible and justifiable for the current application of illustrating retrieval efficacy for various parameters using combined microwave and sub-mm channels. But will a NN approach be untenable for real-time retrievals from space borne sensors (for instance, when ICI and MWI are eventually launched)?**

Answer:

The computational burden is not high. Once the neural networks are trained, which took in our case a few hours, they are very fast making them actually very feasible for real time applications. The main issue of neural networks, as written above, is

that the database covers the range of possible atmospheric conditions. Therefore, the main computation time is needed for creating the database, but this is the issue of all database based retrieval methods as for example Bayesian Monte Carlo Integration. We added a similar statement in Sect. 3.4 (p. 14 lines 11-22).

Reviewer:

**Page 22, lines 5-6: Any specific reason why a 3.5 minute running average was chosen versus a different time averaging duration?**

Answer:

A $3.5$ min running mean corresponds to a path length of $23$ km. This is in the order of the smallest horizontal size of features that can be resolved within of the ICON model, which is twice the grid resolution of ICON. We added a similar statement in Sect. 5 (p. 23 lines 11-12).

Reviewer:

**Section 5.1: So the NN training datasets from all 3 numerical simulations were used for the retrievals shown in this section? Or was the training dataset from the 18 March simulation results applied exclusively to this case? This is not a major issue for the results presented in this particular study, but my main question is how a similar NN retrieval methodology can be applied to a global dataset. Would the training dataset require daily simulations to provide a robust training dataset for global retrievals? Or would a handful of simulations that temporally and strategically sampled various sea- sons suffice?**

Answer:

Yes, the training dataset includes profiles from all three numerical simulations. We added an additional figure (Fig. 1), where the position and time of each randomly selected profile is shown, to emphasize this in Sect 3.1. Neural network retrievals can be easily expanded to global retrievals. The main issue, as written above, is the training database. A training database consisting of similar simulations like our but for tropics, midlatitude and subarctic including the four seasons for midlatitude and subarctic would be probably already enough to provide a reasonably retrieval. Compared to Bayesian Monte Carlo Integration, neural network are also less demanding on the database size (Jiménez et al., 2007).

Reviewer:

**Summary section: As mentioned, ice scattering simulations are rare for sub-mm frequencies. I would add more emphasis in this section (another sentence or two) to encourage the community to produce more ice model scattering datasets at frequen- cies exceeding 200 GHz. This seems like a necessary research step to improve ice and snow column retrievals at sub-mm frequencies.**

Answer:

We followed your suggestion and added some few sentences (p. 32 lines 31-33)

**References**

Gerrit Holl, Salomon Eliasson, Jana Mendrok, and S.A. Buehler. Spare-ice: Synergistic ice wa- ter path from passive operational sensors. *Journal of Geophysical Research: Atmospheres,*
119(3):1504–1523, 2014.

C. Jiménez, S. A. Buehler, B. Rydberg, P. Eriksson, and K. F. Evans. Performance simulations for a submillimetre-wave satellite instrument to measure cloud ice. *Quarterly Journal of the Royal Meteorological Society*, 133(S2):129–149, November 2007. ISSN 1477-870X. doi: 10.1002/qj.134.